



# Geometric Controls of Tidewater Glacier Dynamics

Thomas Frank[1,2], Henning Åkesson[3,4], Basile de Fleurian[1], Mathieu Morlighem[5], and Kerim H. Nisancioglu[1,6]

[1]Department of Earth Science, University of Bergen and Bjerknes Centre for Climate Research, Bergen, Norway
[2]Department of Physical Geography, Stockholm University, Sweden
[3]Department of Geological Sciences, Stockholm University, Sweden
[4]Bolin Centre for Climate Research, Stockholm, Sweden
[5]Department of Earth System Science, University of California, Irvine, USA
[6]Centre for Earth Evolution and Dynamics, University of Oslo, Oslo, Norway

**Correspondence:** Thomas Frank (t.frank95@web.de)

**Abstract.** Retreat of marine outlet glaciers often initiates depletion of inland ice through dynamic adjustments of the upstream glacier. The local topography of a fjord may promote or inhibit such retreat, and therefore fjord geometry constitutes a critical control on ice sheet mass balance. To quantify the processes of ice-topography interactions and enhance the understanding of the dynamics involved, we analyze a multitude of topographic fjord settings and scenarios using the Ice-sheet and Sea-level

System Model (ISSM). We systematically study glacier retreat through a variety of artificial fjord geometries and quantify the modeled dynamics directly in relation to topographic features. We find that retreat in an upstream widening or deepening fjord does not necessarily promote retreat, but conversely, may stabilize a glacier because converging ice flow towards a constriction enhances lateral shear. An upstream narrowing or shoaling fjord, in turn, may promote retreat since fjord walls or bed provide little stability to the glacier where ice flow diverges. Furthermore, we identify distinct quantitative relationships

directly linking grounding line discharge and retreat rate to fjord topography, and transfer these results to a long-term study of the retreat of Jakobshavn Isbræ. These findings offer new perspectives on ice-topography interactions, and give guidance to an ad-hoc assessment of future topographically induced ice loss based on knowledge of the upstream fjord geometry.

## 1   Introduction

Rates of ice discharge from the Greenland Ice Sheet are likely to exceed their Holocene (last 12,000 yrs) maxima this century (Briner et al., 2020; Kajanto et al., 2020), and parts of Antarctica are on the brink of irreversible mass loss (Garbe et al., 2020). Consequently, major natural and societal challenges related to changes in the terrestrial cryosphere of the high latitudes lay ahead. An advanced understanding of the underlying processes of ice loss is paramount for fact-based decision making (Oppenheimer et al., 2019).



About half of the current mass loss over Greenland (30 to 70%) is due to dynamic ice discharge related to thinning, speed-up and increased calving of outlet glaciers (Nick et al., 2009; Felikson et al., 2017; Haubner et al., 2018; Mouginot et al., 2019; King et al., 2020). In Antarctica, dynamic instability of the West Antarctic Ice Sheet is considered a major driver of future sea-level rise (Pattyn and Morlighem, 2020), but there is also emerging evidence of changes in ice dynamics at some glaciers in East Antarctica (Brancato et al., 2020; Miles et al., 2020). While outlet glaciers therefore are critical to ice-sheet mass balance

and associated sea-level rise, considerable knowledge gaps on the processes governing their dynamics still exist.

Despite the general warming trend observed over the recent decades, we do not observe an overall synchronous pattern in outlet glacier evolution. This is clear for various settings in the Arctic, such as in Greenland (Warren and Glasser, 1992; Carr et al., 2013; Bunce et al., 2018; Catania et al., 2018), Svalbard (Schuler et al., 2020), Novaya Zemlya (Carr et al., 2014) and North America (McNabb and Hock, 2014), as well as in Antarctica (Pattyn and Morlighem, 2020). Even adjacent glaciers

with similar climatic and oceanic conditions can show strongly different behaviour (Carr et al., 2013; Catania et al., 2018; Bunce et al., 2018). The main proposed explanation is that differing bathymetry and glacier geometry significantly modulate glacier response to climate over a range of time scales (Warren and Glasser, 1992; Briner et al., 2009; Jamieson et al., 2012; Åkesson et al., 2018a; Catania et al., 2018). Broadly, there is a consensus that wide and deep parts of a fjord promote retreat, while narrow and shallow areas tend to stabilize glacier termini. Moreover, kinematic wave theory indicates that the upstream

propagation of a thinning signal is heavily influenced by bed topography (Felikson et al., 2017, 2021). Modelling of idealized settings (Enderlin et al., 2013; Åkesson et al., 2018b) and theoretical studies based on analytical calculations and numerical experiments further emphasize the potential of fjord geometry to modulate glacier retreat (Weertman, 1974; Raymond, 1996; Vieli et al., 2001; Schoof, 2007; Pfeffer, 2007; Gudmundsson et al., 2012; Gudmundsson, 2013).

It is therefore critical to advance our understanding of the influence of fjord geometry on glacier retreat to accurately predict

sea-level rise, especially when extrapolating observations of a few well-monitored glaciers to those less studied (Nick et al., 2009). This knowledge is also pivotal to correctly infer past climate signals from glacier proximal landforms, because their formation may have been influenced by fjord topography and may not necessarily have been in equilibrium with the prevailing climate (Åkesson et al., 2018b; Steiger et al., 2018).

The most important suggested mechanisms behind geometric control of glacier retreat are: 1) friction, with glaciers in narrow

fjords and grounded well above flotation being stabilized by fjord geometry, while the opposite is the case for glaciers in wide fjords and close to flotation (Raymond, 1996; Pfeffer, 2007; Enderlin et al., 2013; Åkesson et al., 2018b). Buttressing and lateral shear between an ice shelf and nearby islands and/or fjord walls can be an important factor as well (Gudmundsson et al., 2012; Gudmundsson, 2013; Jamieson et al., 2012, 2014); 2) area exposed to ocean melt, where a wider/deeper fjord induces a larger cumulative melt flux (Straneo et al., 2013; Åkesson et al., 2018b); 3) the marine ice sheet instability (MISI), which

is a feedback mechanism between increasing driving stress with increasing ice thickness at the grounding line (GL), where inland-sloping (retrograde) beds lead to self-accelerating ice loss (Weertman, 1974; Schoof, 2007).

While the main controls of ice-topography interactions are known, a quantitative understanding is still largely missing, especially on timescales beyond a few decades. In this context, in-situ observations of ice dynamics do not cover the full spectrum of ice-topography interactions, because they are limited in space and time. While remotely-sensed observations of





ice dynamics over the past decades exist, the recent retreat in Greenland and elsewhere over this period is too short to allow for
a complete assessment of geometry-glacier interactions (Carr et al., 2013; Catania et al., 2018; Bunce et al., 2018). In contrast,
on paleo-time scales, retreat has occurred over large distances, but the temporal resolution of geomorphological studies is
limited by the available geological data and key information is missing to discern different drivers of glacier retreat (Briner
et al., 2009; Åkesson et al., 2020). Meanwhile, numerical studies that can address these issues have so far mostly used width-
and depth-integrated flowline models, which carry many assumptions that do not hold in some settings (Nick et al., 2009;
Enderlin et al., 2013; Åkesson et al., 2018b; Steiger et al., 2018). In particular, they parameterize or do not account for factors
that are thought to be instrumental to explain ice-topography interactions, such as lateral drag, across-flow variations in glacier
characteristics and viscosity changes due to variations in ice temperature. The latter, for instance, was found to be key to explain
Jakobshavn Isbræ's recent retreat (Bondzio et al., 2017).

Here, we use a numerical ice-flow model resolving two horizontal dimensions, we include a larger suite of experiments
than previous related studies, and present a systematic approach to compare the relative importance of basal and lateral fjord
topography. This setup allows to assess how fjord topography controls glacier retreat on interannual to centennial time scales.
We hypothesize that there are quantifiable relationships between glacier retreat and topography that apply to a wide range of
glaciological settings. Such general relationships would yield substantial predictive power for a broad assessment of expected
future outlet glacier retreat.

We create a large ensemble of artificial fjords that include geometric features (referred to as 'perturbations' in the following)
typically found in glacial fjords, such as sills and overdeepenings (referred to as 'bumps' and 'depressions', respectively;
together 'basal perturbations') as well as narrow and wide fjord sections (referred to as 'bottlenecks' and 'embayments',
respectively; together 'lateral perturbations'). We then force synthetic glaciers to retreat through this variety of fjords by
increasing ocean-induced melt rates, and assess key retreat metrics such as the grounding line retreat rate. The ice dynamics
of each simulation are compared and quantitatively linked to the characteristics of the respective fjord geometry. Finally, we
investigate whether the retreat dynamics and ice-topography interactions are transferable from the idealized setup to a long-
term study on Jakobshavn Isbræ in western Greenland.

## 2 Methods

### 2.1 Ice sheet model

We use the Ice-sheet and Sea-level System Model (ISSM; Larour et al., 2012) with the shallow-shelf approximation (SSA;
Morland, 1987; MacAyeal, 1989). Our domain is rectangular (80 km×10 km) with $x$ and $y$ representing the along-flow and
across-flow coordinates, respectively (Fig. 1a). We create an unstructured mesh with a fixed resolution of 100 m close to
the GL, comparable to other high-resolution studies of Greenlandic fjords (e.g. Morlighem et al., 2016, 2019). The temporal
resolution is set to $\Delta t = 0.01$ yrs (3.65 days) to satisfy the Courant-Friedrichs-Levy condition (Courant et al., 1928). We apply a
sub-element GL migration scheme (Seroussi et al., 2014) and enable a moving calving front with the level-set method (Bondzio
et al., 2016). We use a thermal model to account for the feedback between frictional heating and ice viscosity. The spin-up ice





temperature is -5°C, representative of conditions in South/Central Greenland and the Southern Fennoscandian ice sheet (Nick et al., 2013; Åkesson et al., 2018a). To simulate calving, a von-Mises law is used (Morlighem et al., 2016), where the calving rate $c$ is given as:

$$c = ||\boldsymbol{v}|| \frac{\tilde{\sigma}}{\sigma_{max}}, \tag{1}$$

where $\boldsymbol{v}$ is the velocity vector, $\tilde{\sigma}$ a scalar quantity representing the tensile stress at the ice front, and $\sigma_{max}$ is a stress threshold. This formulation, demonstrated to perform well in Greenlandic fjords (Choi et al., 2018), means that calving occurs when the tensile stress at the glacier front exceeds a fixed threshold. $\sigma_{max}$ generally needs to be determined experimentally for studies on real-world glaciers. Here we fix $\sigma_{max}$ to 1 MPa for grounded ice and 200 kPa for floating ice. These values yield a representative setup and are within the range suggested for Greenland outlet glaciers (Morlighem et al., 2016; Choi et al., 2018, 2021). Basal sliding is parameterized with a Budd type friction law (Budd et al., 1984) of the form

$$\boldsymbol{\tau}_b = -k^2 N \boldsymbol{v}_b, \tag{2}$$

where $\boldsymbol{\tau}_b$ is basal drag, $k$ is a friction parameter, $\boldsymbol{v}_b$ is the the basal velocity, and $N$ is the effective pressure. $N$ is given as $N = \rho_i g H - \rho_w g \max(0, -z_B)$ where $\rho_i$ and $\rho_w$ are the density of ice and salt water, respectively, $g$ is gravitational acceleration, $H$ is ice thickness and $z_B$ is bed elevation with respect to sea level. $N$ is thus the difference between the ice overburden and water pressure assuming perfect connectivity between the subglacial water layer and the ocean. We set $k$ spatially uniform to isolate the topographic signal of retreat in our results and thus to reduce the number of degrees of freedom for the interpretation. We fix $k = 40$ (Pa yr m$^{-1}$)$^{\frac{1}{2}}$, which is mid-range among values typically found in glaciological settings resembling ours (Bondzio et al., 2017; Haubner et al., 2018; Åkesson et al., 2018a).

Melting under floating ice is parameterized through prescribed melt rates that are invariant of depth. The reference forcing used for model spin-up is a subshelf melt rate of 30 m yr$^{-1}$ and a frontal rate of undercutting of 200 m yr$^{-1}$, where the latter acts on the vertical ice front if the terminus is grounded. Both values are on the lower end of observed melt rates (Motyka et al., 2003; Enderlin and Howat, 2013; Xu et al., 2013), thus reflecting a configuration prior to recent warming when glaciers were more in balance with the ambient climate than today (King et al., 2020).

As part of the idealized setup, the surface mass balance (SMB) is fixed to zero, except in the uppermost 10 km of the domain (Fig. 1a), where we set an accumulation rate of 55 m yr$^{-1}$. This creates a realistic fixed upstream ice flux and is not meant to represent local SMB found on real glaciers. Additionally, mass is added to the model domain by fixing an ice velocity of $v_x = 50$ m yr$^{-1}$ at the upper domain boundary, which creates an influx as a function of the glacier thickness. These two approaches to adding mass represent a total accumulation ($A$) which can be expressed as

$$A = C + \int_0^{y_{max}} v_x(0, y) H(0, y) \, dy, \tag{3}$$

where $C = \left(10 \times 10^3\right)^2 \times 55 = 5.5 \times 10^9$ m$^3$ yr$^{-1}$ is a constant accumulation. Through the thickness-dependent influx represented by the second term on the right-hand side, we account for a reduction in accumulation for a shrinking glacier, thus parameterizing the SMB - altitude feedback (Harrison et al., 2001; Oerlemans and Nick, 2005).

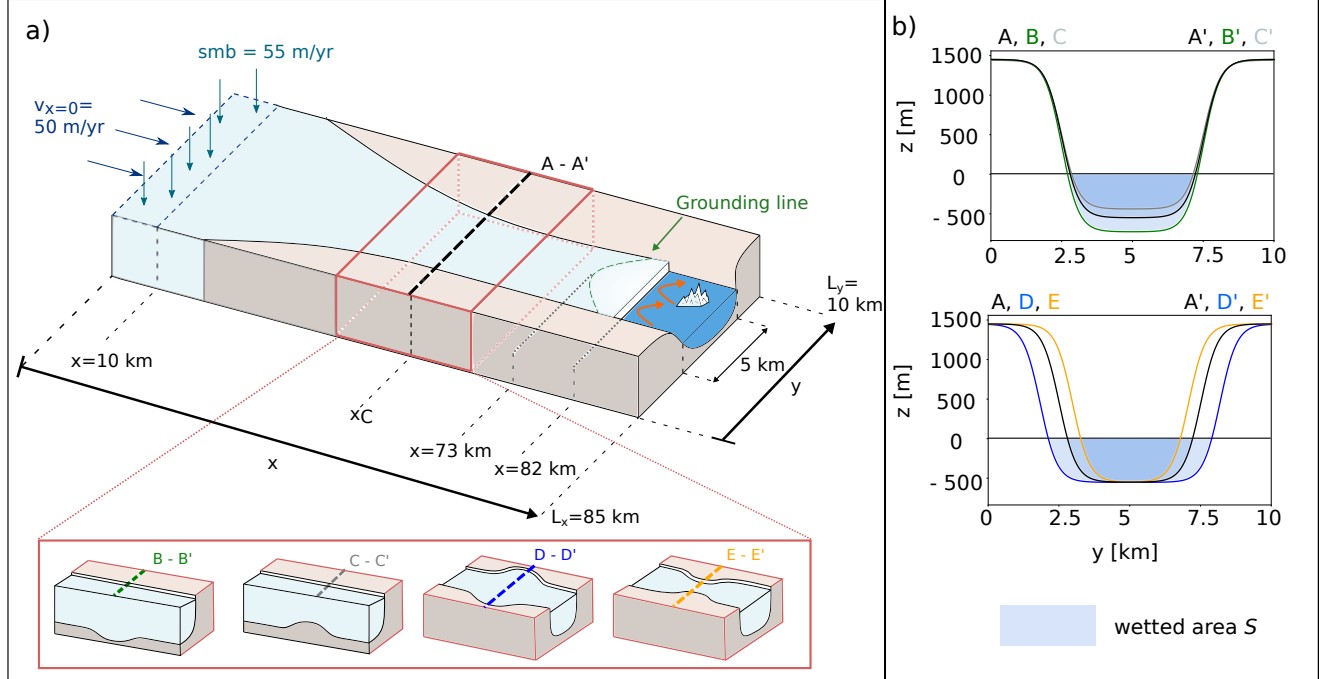

**Figure 1.** Schematic of the experimental setup. a) Sketch of the domain (not to scale) with annotated dimensions and mass balance processes (gains: thickness-dependent influx and surface accumulation; losses: melt at the ice-ocean interface and calving). Red box symbolizes how the fjord geometry is changed in different experiments to include geometric perturbations (their center being referred to as $x_C$). b) Cross-sections through the linear fjord (black line) and geometric perturbations. Upper panel: basal perturbations (green: depression, grey: bump); Lower panel: lateral perturbations (blue: embayment, yellow: bottleneck). The wetted area, i.e. the cross-sectional area of the fjord below sea level, is shaded in blue for each geometry.

We impose free-slip boundaries at the lateral margins of the domain (where $y = 0$ and $y = y_{max}$), meaning that no mass can leave the system laterally. In summary, the only mass source is at the upstream end of the domain, while the only mass wasting occurs where the glacier is in contact with the ocean (either through calving or melting).

## 2.2 Fjord Geometries

Our reference geometry is a fjord sloping linearly towards the ocean with a wide section in the upstream area from which ice is
funneled towards a 5 km wide outlet channel with parallel walls (Fig. 1a). The fjord topography is given by $B(x, y) = B_x(x) + B_y(x, y)$ with

$$B_x(x) = B_0 + x \times \alpha + \Theta(x) \tag{4}$$



**Table 1.** Parameters for generating fjord geometries (in parentheses for *longer* geometric perturbations).

| parameter | value | unit | description |
|---|---|---|---|
| $B$ | | m | bed elevation |
| $B_0$ | -450 | m | bed elevation at x=0 |
| $\alpha$ | -0.002 | | slope of bed in x-direction |
| $d_f$ | 2000 | m | depth of fjord relative to upland areas on the sides |
| $L_y$ | 10 | km | width of domain in y-direction |
| $L_x$ | 85 | km | length of domain in x-direction |
| $w_f$ | 2.5 | km | half-width of fjord |
| $m_f$ | $\frac{1}{300}$ | | factor for steepness of fjord walls |
| $x_U$ | 30 | km | extent in x-direction of wide upstream area |
| $F$ | 300 | | factor for smooth transition between wide upstream area and parallel fjord |
| $x_B$ | 45 (35) | km | x-coordinate of upstream end of perturbation |
| $x_E$ | 65 (65) | km | x-coordinate of downstream end of perturbation |
| $x_C$ | 55 (50) | km | x-coordinate of center of perturbation |
| $\Lambda$ | 20 (30) | km | horizontal extent of perturbation in x-direction |
| $\Gamma$ | *variable* | m | deviation of fjord half-width or depth relative to parallel fjord at $x_C$ |

and

$$B_y(x,y) = \frac{d_f - B_x(x)}{1 + e^{\left(m_f\left(y - \frac{1}{2}L_y + w_f + \Omega(x) + \Theta(x)\right)\right)}} + \frac{d_f - B_x(x)}{1 + e^{\left(-m_f\left(y - \frac{1}{2}L_y - w_f - \Omega(x) - \Theta(x)\right)\right)}}, \tag{5}$$

where $\Omega(0 < x < x_U) = \frac{x - x_U}{F}^2$. The parameter values and descriptions are found in Table 1. This formulation is inspired by the MISMIP setup (Gudmundsson et al., 2012; Asay-Davis et al., 2016), but adapted to our purpose.

To insert basal or lateral perturbations in the outlet channel and thus alter the fjords' depth or width in specific areas, we modify the parameter $\Theta(x)$ in either eq. 4 or eq. 5 such that

$$\Theta(x_B < x < x_E) = -\sin\left(\frac{2\pi}{\Lambda}\left(x - \frac{\Lambda}{4} - x_C\right)\right)\frac{\Gamma}{2} + \frac{\Gamma}{2}. \tag{6}$$

Altering $\Theta(x)$ only in one of the terms on the right-hand side of eq. 5 allows to produce fjords with one-sided lateral perturbations, thus making them asymmetric. Combined, these equations reproduce the typical U-shape of fjords (Fig. 1b) and yield a setting that is representative of a wide range of outlet glaciers.

The metric used to quantitatively link fjord shape with glacier retreat is the wetted area $S$: the submerged cross-sectional area of the fjord (Fig. 1b), which can be calculated at every point along an outlet channel. This metric combines information
about the width and depth of a fjord and is thus a comprehensible parameter for comparing basal and lateral perturbations.





**Table 2.** Suite of experiments with name (extensions _lon and _asy refer to *longer* and *asymmetric* geometries), type of geometric perturbation, perturbation magnitude, the deviation of fjord width (two times $\Gamma$ for symmetric lateral perturbations, one time $\Gamma$ for asymmetric ones) or depth (one time $\Gamma$ for basal perturbations) at the center of the perturbation relative to the linear reference fjord, $S$ at $x_C$ (i.e. the wetted area at the center of the perturbation) and forcings required to induce complete retreat through the entire geometric perturbation (/ if no complete retreat could be enforced).

| Experiment | Perturbation type | Perturbation magnitude | Fjord width/depth deviation [m] | $S$ at $x_C$ [km$^2$] | Forcing for complete retreat (undercutting/subshelf melt rate) [m yr$^{-1}$] |
|---|---|---|---|---|---|
| Ref | - | - | 0 | 2.1 | 800/120 |
| ByH450 | embayment | small | 900 | 2.6 | 1200/180 |
| ByH900 | embayment | medium | 1800 | 3.1 | 1200/180 |
| ByH1350 | embayment | large | 2700 | 3.6 | 1200/180 |
| ByH-450 | bottleneck | small | -900 | 1.6 | 800/120 |
| ByH-675 | bottleneck | medium | -1350 | 1.3 | / |
| ByH-900 | bottleneck | large | -1800 | 1.1 | / |
| BuH-120 | depression | small | -120 | 2.6 | 1200/180 |
| BuH-240 | depression | medium | -240 | 3.1 | 1000/150 |
| BuH-360 | depression | large | -360 | 3.6 | 800/120 |
| BuH120 | bump | small | 120 | 1.6 | / |
| BuH180 | bump | medium | 180 | 1.3 | / |
| BuH240 | bump | large | 240 | 1.1 | / |
| ByH900_lon | embayment | medium | 1800 | 2.8 | 1200/180 |
| ByH-675_lon | bottleneck | medium | -1350 | 1.5 | 1200/180 |
| BuH-240_lon | depression | medium | -240 | 2.8 | 1200/180 |
| BuH180_lon | bump | medium | 180 | 1.5 | / |
| ByH900_asy | embayment | small | 900 | 2.6 | 1200/180 |
| ByH1800_asy | embayment | medium | 1800 | 3.1 | 1200/180 |
| ByH-900_asy | bottleneck | small | -900 | 1.6 | 1200/180 |
| ByH-1800_asy | bottleneck | large | -1800 | 1.1 | / |





Furthermore, it is straightforward to calculate its first derivative, $dS$, which yields information on the along-fjord change in width or depth.

Besides our reference setup, we test 20 fjord geometries (Table 2), each of which contains either a small, medium or large geometric perturbation. The magnitude of the perturbation is defined by how much the width or depth of the fjord deviates from the reference fjord. Our 'core experiment', which the results will focus on, comprises 12 fjords, each of which features one of the four perturbation types (depression, bump, bottleneck, embayment) of one of the three magnitude classes (small, medium, large). The depressions and embayments in each magnitude class increase the wetted area $S$ at the center of the perturbation $x_C$ by the same amount, while the bottlenecks and bumps in each magnitude class reduce $S$ at $x_C$ by the same amount. The along-flow horizontal extent of all perturbations in the core experiment is 20 km (Fig. 2).

Additionally, we test *asymmetric* and *longer* perturbations to verify if the results from our core experiment can be transferred to a wider range of settings. We test two asymmetric embayments, which have the same $S$ at $x_C$ as the small and medium embayments and depressions, as well as two asymmetric bottlenecks which have the same $S$ at $x_C$ as the small and large bottlenecks and bumps. The longer perturbations have an along-flow horizontal extent of 30 km. We test one longer perturbation per perturbation type with $S$ at $x_C$ corresponding to the medium magnitude class.

## 2.3 Reference Glacier

All experiments start from an artificial reference glacier, which is produced by relaxing a rectangular block of ice in the reference fjord. The spin-up is run until the relative ice volume $((dV/yr)/V << 0.05\%)$ and GL position are stable. The length of the spun-up reference glacier is 82 km, its GL is at $x = 73$ km and the velocity at the GL along the central flow line of the glacier $v_{GL} = 2.5$ km yr$^{-1}$. At steady-state, the total mass gain is $\sim 6.1$ km$^3$ yr$^{-1}$ ($\approx 5.6$ Gt yr$^{-1}$), which is balanced by mass losses through melting at the ice - ocean interface ($\sim 0.9$ km$^3$ yr$^{-1}$) and calving ($\sim 5.2$ km$^3$ yr$^{-1}$). In a sensitivity experiment with doubled ocean melt rates (400 m yr$^{-1}$ undercutting and 60 m yr$^{-1}$ subshelf melting), mass loss through ocean melt increases to $\sim 2.1$ km$^3$ yr$^{-1}$, while calving reduces to $\sim 4$ km$^3$ yr$^{-1}$. The GL remains largely stable, indicating that the reference glacier is not very sensitive to ocean forcing due to compensating effects in the mass wasting processes.

The setup represents a medium sized fjord-glacier system, which has similar dimensions and dynamics as, for example, the present-day Alison glacier in NW Greenland, where the fjord width is about 5 km, water depth is around 500 m and observed ice discharge has increased from $\sim 4$ to $\sim 8$ Gt yr$^{-1}$ in the past 20 yrs (Mouginot et al., 2019). It is furthermore broadly representative of outlet glaciers from the Fennoscandian ice sheet during the last glacial, such as the Hardangerfjorden glacier (Mangerud et al., 2013; Åkesson et al., 2020).

## 2.4 Retreat Experiments in variable fjords

We slightly modify the reference glacier to match the new fjord geometry when introducing geometric perturbations. For embayments, we extrapolate the glacier surface laterally to fill the newly introduced lateral cavities. For depressions, we fill the new basal cavity with ice, but keep the glacier surface the same. For bumps or bottlenecks, we remove ice while keeping

**Figure 2.** Along-fjord profiles of the wetted area $S$ (a: lateral; b: basal perturbations) and its derivative $dS$ (c: lateral; d: basal) for fjords featuring different geometric perturbations of different magnitude classes. Note that the profiles of embayments and depressions, and likewise, bottlenecks and bumps, of the same magnitude class are largely congruent, thus allowing a straightforward comparison between basal and lateral perturbations.





the glacier surface unaltered. Subsequently, we relax the glacier in each geometry for 50 yrs, resulting in an ice volume change $(dV/yr)/V < 0.5\%$ at the end of relaxation for every setup tested and a stable GL.

After relaxation, we increase the ocean forcing to trigger a retreat. We aim to force the GL to retreat through the entire geometric perturbations. The ocean melt rates required to induce such a retreat depend on the fjord geometry, which we elaborate on in the results section. To determine what melt rates are needed to force this complete retreat in a particular fjord, we strengthen the ocean forcing using multiples of the reference forcing (200 m yr$^{-1}$ frontal rate of undercutting, 30 m yr$^{-1}$ subshelf melt) until complete retreat takes place. In some cases (c.f. sect. 3.2), even unrealistically high values for the ocean

forcing (e.g. 20 times the reference forcing) did not trigger complete retreat, suggesting that glaciers in these geometries are not sensitive to ocean melt.

Since we want to explore the response of outlet glaciers to melting at the ice-ocean interface, we keep the SMB constant with time, and let the upstream ice flux vary only through our parameterized SMB - altitude feedback (Eq. 3).

We assess 16 glacier metrics during the retreat, which we expect to show a response to local topography (Table 3). All of

these can be observed in-situ or via remote sensing techniques (e.g. Mouginot et al., 2019; King et al., 2020), which means that our results are readily transferable to real-world settings. The GL position ($x_{GL}$) and its derivative, the GL retreat rate ($dGL$), the front position ($x_{Fr}$) and its derivative, the frontal retreat rate ($dFr$), but also the velocity at the GL ($v_{GL}$) and the shelf length ($L_S$) are measured along the central flow-line of the glacier.

### 2.5    A real-world case study: Jakobshavn Isbræ

We want to verify to what degree the dynamics seen in our experiments are also prevalent in real-world settings. This is challenging since we investigate decadal to centennial time scales. Specifically, we would need observations with high temporal resolution on glacier metrics (Table 3) for a glacier that has retreated over tens of kilometers, through a fjord with variable and known topography. There are perhaps only a handful of glaciers worldwide that may fulfil these requirements, and even so, acquiring the necessary data is difficult and outside the scope of the present study.

To test our idealized results in a real-world setting, we instead turn to model simulations from Jakobshavns Isbrae's (JI) evolution in the Holocene (Kajanto et al., 2020). We focus on a simulation of the retreat of JI from a sill at the fjord mouth of Jakobshavn Isfjord, to a point inland of today's GL position. This model retreat is forced using a step reduction in the equilibrium line altitude early in the Holocene (experiment SE_CM in Kajanto et al. (2020)). While this is a sensitivity experiment not meant to reflect the actual evolution of JI (Kajanto et al., 2020), it is convenient for our purposes since it produces a

long-lasting, dynamic retreat.

Just like in our idealized experiments, we calculate *S* and *dS* for Jakobshavn Isfjord. We then asses whether the relationships found in our idealized settings are also prevalent in JI's dynamics (sect. 3.5).



**Table 3.** Glacier characteristics assessed during retreat for later correlation with fjord geometry. Parameters marked with * are assessed along the central flow-line of the glacier.

| Glacier metric | Variable | Unit |
|---|---|---|
| Grounding line position* | $x_{GL}$ | km |
| Grounding line retreat rate* | $dGL$ | m yr$^{-1}$ |
| Front position* | $x_{Fr}$ | km |
| Front retreat rate* | $dFr$ | m yr$^{-1}$ |
| Grounding line mass flux | $Q_{GL}$ | km$^3$ yr$^{-1}$ |
| Ice front mass flux | $Q_{Fr}$ | km$^3$ yr$^{-1}$ |
| Flux through an upstream gate | $Q_U$ | km$^3$ yr$^{-1}$ |
| Calving Flux | $Q_C$ | km$^3$ yr$^{-1}$ |
| Velocity at the grounding line* | $v_{GL}$ | m yr$^{-1}$ |
| Maximum velocity | $v_{max}$ | m yr$^{-1}$ |
| Shelf length* | $L_S$ | m |
| Floating Area | $A_F$ | m$^2$ |
| Grounded Area | $A_G$ | m$^2$ |
| Ice Volume | $V$ | km$^3$ |
| Ice Volume Above Flotation | $V_{AF}$ | km$^3$ |
| Maximum Ice Thickness | $H_{max}$ | m |

# 3 Results

## 3.1 Stable and Unstable Grounding Line Positions

We identify GL positions of intermittent stability ('stable' GL positions), i.e. where the GL rests for a sustained time (typically 50 to 200 yrs), or retreats very slowly, and areas where the GL retreats quickly ('unstable' GL positions). Fig. 3 shows both stable and unstable positions for one representative run per perturbation type. Note that in the following, all terminology related to along-fjord changes in width or depth of the fjord (e.g. narrowing, deepening) will refer to the direction of glacier retreat.

   Stable positions exist at the downstream end of embayments and depressions where the fjord becomes wider/deeper ($x \approx$ 210   62 to 65 km, Fig. 3a,c). Unstable positions, associated with rapid retreat, are found in the remainder of both perturbations ($x \approx$ 45 to 62 km). Retreat from the stable position at the downstream end of the embayment occurs gradually ($dGL \approx -22$ m yr$^{-1}$ while $x_{GL} \approx 62$ to 64 km), but is rapidly accelerating as the GL retreats further into the perturbation, accompanied by some lateral ungrounding. Retreat from the stable position in the depression occurs suddenly after a phase of near-stagnation ($dGL \approx -6$ m yr$^{-1}$, while $x_{GL} \approx 64.5$ to 65.5 km) as the glacier ungrounds where the fjord is deepest in the center of the perturbation 215   ($x \approx 55$ km, Fig. 3c inset plot). The cavity formed under the glacier rapidly grows in size and expands downstream until it





eventually detaches the glacier from the bed also at the downstream end of the depression ($x \approx 65$ km). In bottlenecks and on bumps, stable positions are found where the fjord is narrow/shallow ($x \approx 55$ to $58$ km, Fig. 3b,d). The stabilizing effect of bumps is, in fact, so large that no glacier could be forced to retreat over them within reasonable limits for the ocean forcing. However, we observe that retreat onto bumps occurs fast ($dGL \approx -500$ m yr$^{-1}$ for $x_{GL} \approx 58$ to $65$ km). For bottlenecks, only

the glacier situated in the fjord with a 'small' bottleneck (i.e. the bottleneck with the largest $S$ among the ones tested) could be forced to retreat completely. Noticeably, retreat at the downstream end of the bottleneck ($x \approx 56$ to $65$ km), where the fjord narrows in, is fast (unstable) with $dGL \approx -900$ m yr$^{-1}$, whereas it is very slow (relatively stable) with $dGL \approx -25$ m yr$^{-1}$ upstream of the narrowest point, where the fjord is widening ($x \approx 45$ to $55$ km; Fig. 3d).

In summary, relatively stable GL positions are found where the fjord widens/deepens in the direction of glacier retreat

(positive $dS$, Fig. 2) and rapid retreat occurs through areas where the fjord becomes narrower/shallower (negative $dS$). Thus, the along-fjord change in width or depth ($dS$) is a key control on GL retreat. However, in fjords that have a smaller $S$ at $x_C$ than the reference fjord (bottlenecks and bumps), stable positions are also found where the fjord is narrow or shallow (small $S$). Therefore, $S$ is also an important control on GL retreat. The experiments with asymmetric and longer perturbations confirm these findings (see supp. Fig. A1).

## 3.2   Forcings and timings of retreat

Now, we investigate how retreat from stable and unstable GL positions is correlated with fjord topography (i.e. $S$ and $dS$). Two parameters are important in this context (Fig. 4): First, the amplitude of the forcings needed to induce complete retreat through the different geometric perturbations. As mentioned previously, distinctions exist both between the different perturbation types (bumps, depressions, bottlenecks, embayments) as well as the different magnitude classes (small, medium, large) for a given

geometry type. Second, the approximate residence time of the GL in a stable position. The stronger the GL is stabilized by a particular geometric perturbation, the longer it will rest in a stable position.

All glaciers in embayments require the same increase in forcing to retreat completely (6 times the spin-up forcing). This increase is larger than what is needed to induce retreat through the linear reference fjord (4 times the spin-up forcing). The residence time of the GL in the stable positions at the downstream end of the embayments are such that the glacier in the

smallest embayment is the earliest to retreat (after 61 yrs of intermittent GL stability), and the one in the largest the latest (after 173 yrs) (Fig. 4). This implies that the larger the embayment, the more stability it provides to the glacier at its downstream end, before retreat through the entire perturbation is possible. A larger embayment means a locally larger along-flow change in wetted area $dS$ at its downstream end (Fig. 2). Thus, there is a positive correlation between GL stability and $dS$. This indicates that $dS$ not only determines the location of stable GL positions in embayments, as shown before, but that it also quantitatively

impacts how stable the GL is.

The glaciers in fjords with depressions require different forcings to retreat completely (small: 6× the reference forcing, medium: 5×, large: 4×). The residence time also varies; retreat over small depressions occurs ∼65 yrs later than over medium and large depressions, which retreat after about the same time (after 169 and 170 yrs, respectively). These findings indicate that the stabilizing effect of a depression declines the deeper it is (Fig. 4). Thus, there is a negative correlation between GL

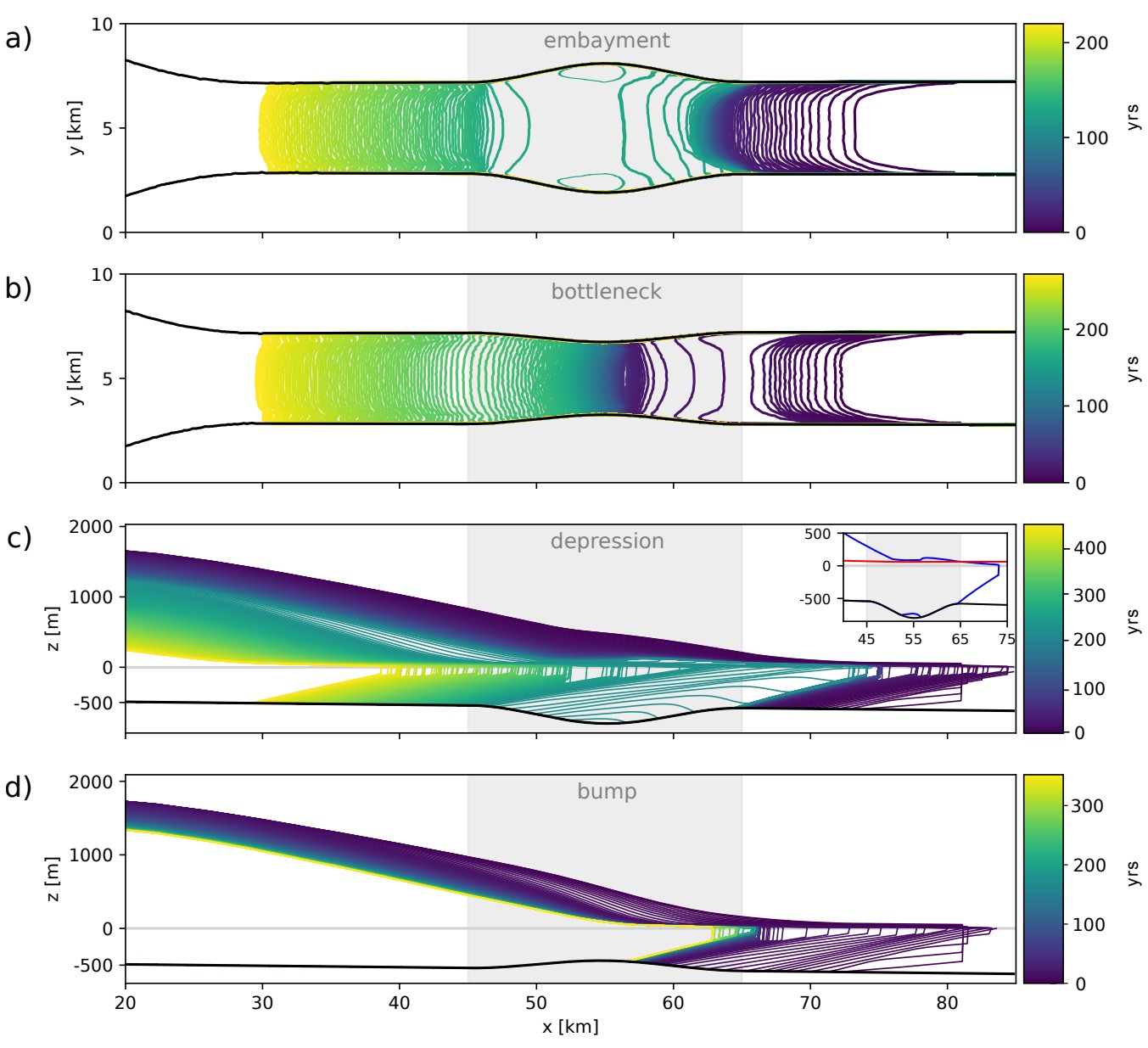

**Figure 3.** Annual glacier evolution (a, b: top down view on domain showing yearly grounding lines; c, d: yearly glacier profiles along central flow line) in fjords featuring different geometric perturbations (location of perturbations marked in grey): a) medium embayment; b) small bottleneck; c) medium depression; d) small bump. Inset plot in c) shows profile (blue) in year 217 when glacier ungrounds in the central part of the depression which triggers further retreat. Red line is the level to which glacier needs to thin to reach flotation.



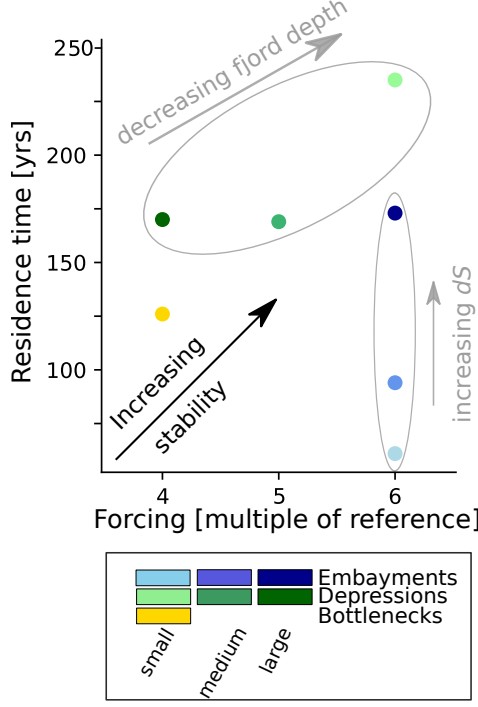

**Figure 4.** Forcing required to induce complete retreat in multiples of the reference forcing (200 m yr$^{-1}$ undercutting rate, 30 m yr$^{-1}$ subshelf melt) and approximate residence time of the GL in an intermittently stable position for different fjord geometries. A longer residence time and a larger forcing required indicate that fjord geometry provides larger stability. More stability is correlated with decreasing fjord depth for depressions (shades of green), and with increasing along-fjord change in wetted area $dS$ for embayments (shades of blue). The simulations for which no retreat through the perturbation was observed have been omitted from the figure.

stability and $S$. In sect. 3.1, ungrounding in the central part of depressions was identified as the trigger of rapid retreat from the temporary stillstands. Such GL retreat occurs more easily in a deeper fjord (larger $S$). Therefore, it is consequential that a deeper depression is less stabilizing. Note, however, that the fjord depth several kilometers upstream of the GL determines for how long the glacier is stabilized. There is no direct correlation between $S$ or $dS$ at the GL and the stability provided to the glacier by the fjord in our settings with depressions.

The glacier in a fjord with a 'small' bottleneck required a four-fold increase in oceanic melt rates, and retreated from its stable position after 126 yrs of intermittent GL stability. This is a weaker forcing than for the glaciers in the embayments, as well as for the medium and small depressions, and thus suggests that this bottleneck provides less stability than these geometries. This contrasts with the common pattern, where a small $S$ and a positive $dS$ should stabilize the glacier strongly. It is unclear why this is the case here. We hypothesize that it might be related to a combination of high driving stresses due to a steepened surface inside the bottleneck in conjunction with high modeled calving rates (not shown). The two experiments with glaciers in geometries with narrower bottlenecks ('medium' and 'large' bottleneck) did not retreat through the entire perturbation.





This, in turn, aligns well with the general notion of a confined (low $S$) and downstream narrowing (positive $dS$) fjord yielding strong stability to the glacier. Likewise, none of the glaciers in fjords with bumps retreated completely, which follows the same concept.

## 3.3 Stress balance response to fjord geometry

To assess the underlying mechanisms behind the geometric controls described before, we now analyze the stress regimes across the studied geometries. We focus on lateral shear stresses for lateral perturbations, and longitudinal stress gradients for basal perturbations as given by the SSA in x-direction by

$$\rho g H \alpha_x = \tau_{bx} + \frac{\partial}{\partial x}(2H\sigma'_{xx} + H\sigma'_{yy}) + \frac{\partial}{\partial y}(H\sigma'_{xy}) \qquad (7)$$

where $\sigma'$ is the deviatoric stress and $\tau_b$ is basal drag. We interpret the second and third term on the right hand side as longitudinal stress gradient and lateral shear, respectively. With the imposed spatially uniform friction coefficient, variations in the investigated stress fields are largely caused by variable fjord topography, and are hence convenient to investigate for our purpose.

For embayments and bottlenecks, variations in lateral shear can be seen along the fjord walls (Fig. 5a,b). Specifically, strongly negative shear stresses are found where ice is funneled in a downstream narrowing fjord. This occurs, for example, where 55 km $< x <$65 km in embayments (Fig. 5a) and at 45 km $< x <$55 km in bottlenecks (Fig. 5b). This indicates enhanced resistance to flow for the glacier originating from the fjord walls, which provides stability to the glacier. Where ice flow diverges in a widening section of the fjord (in the upstream half of embayments and the downstream half of bottlenecks, Fig 5a,b), lateral shear stress is comparatively weak. This indicates that the glacier - fjord wall contact is reduced here, and that the fjord walls provide little support to the glacier in these areas.

For depressions and bumps, we see clear variations in longitudinal stress gradient along the glacier bed (Fig. 5c,d). In depressions, a band of negative values stretches across the full width of the outlet channel where the bed turns from being prograde to retrograde (at $x = 55$ km, Fig. 5c), indicating that ice flow is being blocked here. Likewise, a marked reduction in positive longitudinal stresses is seen at the onset of the bump where the bed slope switches sign (at $x = 45$ km, Fig. 5d). Together, the stress regimes in basal perturbations demonstrate that a retrograde glacier bed, tilted against the direction of flow, reduces longitudinal stress gradients considerably as it increases the basal resistance to flow, which ultimately stabilizes the glacier.

In summary, the stress analysis above suggests that increased lateral shear or negative longitudinal stress gradients are found wherever ice flow is forced to converge, either horizontally or vertically, towards a narrowing or shoaling area downstream. Simulations using asymmetric as well as longer perturbations confirm that these findings are robust (see supp. Fig. B1). Through the convergent flow, the contact between the glacier and the fjord is enhanced, leading to increased resistance to flow. Overall, along-flow change in fjord width or depth (i.e. $dS$) is found to define areas of increased lateral shear or negative longitudinal stress gradients, and thus GL stability.





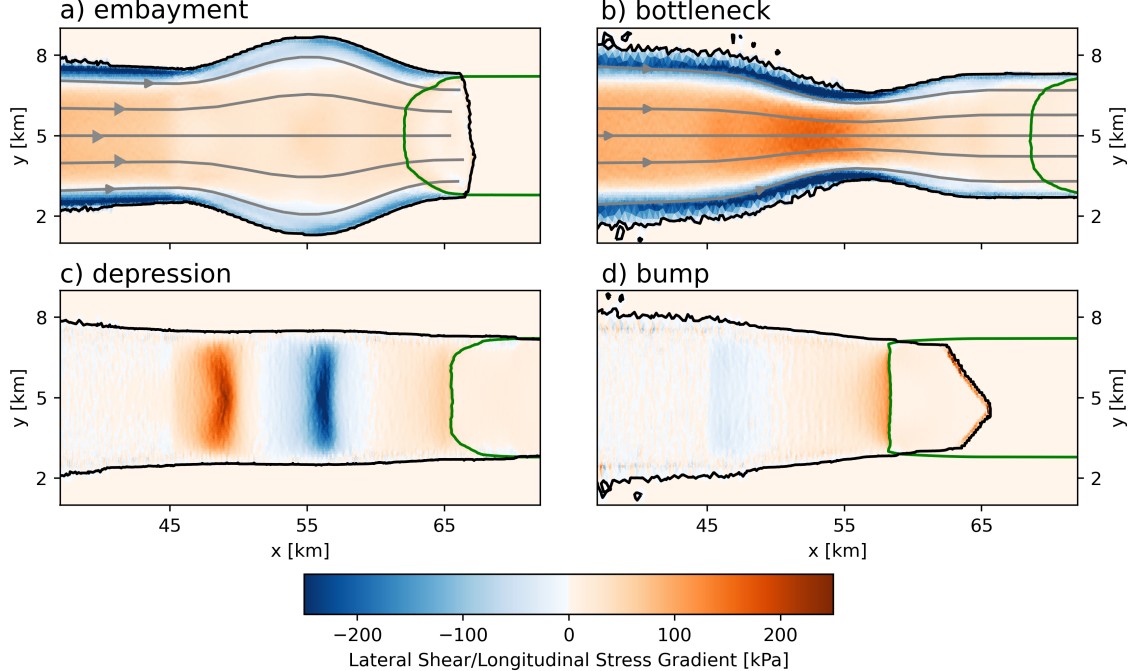

**Figure 5.** Stress states in perturbations. Lateral shear stress for a, b with flow lines in grey, and longitudinal stress gradients for basal perturbations. Green line is grounding line, black line is glacier outline.

## 3.4 A universal quantitative relationship for ice-topography interaction

We hypothesize that there is a quantitative relationship between fjord geometry and glacier retreat, valid across a range of different geometries. To test this, we correlate a variety of metrics indicative of glacier retreat (Table 3) against relevant metrics of fjord geometry, that is, the submerged cross-sectional area ($S$) and its derivative ($dS$). We restrict the data to those instances when the GL is located within a geometric perturbation (gray-shaded areas in Fig. 3). Among all combinations of retreat and geometry metrics tested, including those of asymmetric and 'longer' perturbations (Table 2), the clearest and most

universal relationship found is a negative, close to linear correlation between the ratio of the GL flux and the submerged cross-sectional area $Q_{GL}/S$ over the change in submerged cross-sectional area $dS$ (Fig. 6). This relation expresses that a widening or deepening fjord in the downstream direction (negative $dS$) promotes a high GL flux per wetted area ($Q_{GL}/S$). Conversely, a glacier retreating in a fjord that becomes narrower or shallower downstream (positive $dS$), will have a reduced $Q_{GL}/S$. Note that the ratio $Q_{GL}/S$ for basal geometry perturbations (grey to black and green colors in Fig. 6) is on average lower than for

lateral geometry perturbations. This means that basal perturbations generally inhibit ice flux across the GL more efficiently than lateral perturbations. Also, note that the GL flux is the product of the velocity $v_{GL}$ and the flux gate area at the GL $A_{GL}$, that





is $Q_{GL} = v_{GL} \times A_{GL}$. The ratio $Q_{GL}/S$ is thus proportional to $v_{GL}$ when there is hydrostatic equilibrium at the GL (because in that case, $S = 0.9 \times A_{GL}$), and so we find a comparable, negative linear relationship between $v_{GL}$ and $dS$ (Fig. SC1).

We find an additional, yet less distinct, negative relationship between the wetted area $S$ and the GL retreat rate $dGL$ (Fig. 7a). This shows that a wider or deeper fjord promotes faster GL retreat, while a narrower or shallower geometry stabilizes the glacier. This relation is not as universal as the previous one since one value for $dGL$ is not uniquely linked to one value for $S$ across different geometries. Furthermore, it is not linear, but rather such that for a range of low $S$ values, $dGL$ does not vary noticeably. Only above a certain threshold in $S$, the GL retreats markedly faster (Fig. 7b). This threshold varies between different fjord geometries. However, we find that it is always associated with the location of intermittent GL stability (Fig 3.1).

This means that a relationship between GL retreat rate $dGL$ and $S$ only unfolds if a local stability position is passed. These stable positions can be either where $S$ is low, or where $dS$ is high, as shown previously. For instance, $dGL$ does not increase as the GL retreats very slowly at the stable position in the downstream half of embayments. Only once it has retreated passed this point of intermittent GL stability, a correlation between $dGL$ and $S$ can be seen.

    For depressions we do not see a distinct relationship between $dGL$ and $S$ (Fig. SD1). This is because we measure the GL

position $x_{GL}$ and therefore also $dGL$ as the furthest downstream grounded point along the central flow-line of the glacier. When the glacier ungrounds in the center of a depression, where the fjord is deepest, the dynamics of retreat are triggered several kilometers upstream of the GL, as mentioned in sect. 3.1. Therefore, there is a correlation between fjord depth and GL retreat in depressions. However, it is not reflected when only considering processes at the GL. Not finding a $dGL$ over $S$ relation for depressions is hence expected by construction of our methodology, and not an actual feature.

**3.5   Jakobshavn Isbræ**

    Given our previous results, we now aim to assess whether our principal geometric relationship $Q_{GL}/S$ over $dS$ can be found for Jakobshavn Isbræ. To this end, we calculate the wetted area $S$ along the topography of Jakobshavn Isfjord as used in Kajanto et al. (2020) which depicts overall higher values and larger along-fjord changes in $dS$ than our idealized settings (Fig. 8b).

    Plotting all available data points for $Q_{GL}/S$ over $dS$ at Jakobshavn, we do not find the aforementioned geometric relation-

ship. This may have many reasons related to the complex dynamics of Jakobshavn Isbræ (Bondzio et al., 2017), but most critically, there is lateral inflow of ice to the main channel from the surrounding ice sheet and tributaries (compare Steiger et al., 2018). This alters the stress balance at the GL compared to our experiments where the glacier is always closely confined between fjord walls. Specifically, it means that the rigid glacier-wall interface in our experiments is replaced by a changing ice-ice contact. This has implications for the lateral friction that the fast-flowing ice in the main channel experiences and for

the processes transferring stabilizing back stress from the sides to the center of flow.

    We thus expect our findings to be more easily transferable to settings where Jakobshavn Isbræ is enclosed by fjord walls. This is only the case in one part of the outlet channel, upstream of the present-day front (Fig. 8c). Indeed, values for $Q_{GL}/S$ are inversely related to $dS$ in a qualitative way here, such that an increase in $dS$ is generally associated with a decrease in $Q_{GL}/S$ and vice versa (Fig. 8d,e), consistent with our findings for synthetic geometries above (Fig. 6). Given the complexity

of Jakobshavn Isbræ's dynamics, we find these results encouraging. For settings resembling our setup more closely, such as



**Figure 6.** Relationship between grounding line discharge per wetted area $Q_{GL}/S$ and along-fjord change in wetted area $dS$ for all tested geometries and all instances when the GL is within a geometric perturbation (grey area in Fig. 3).





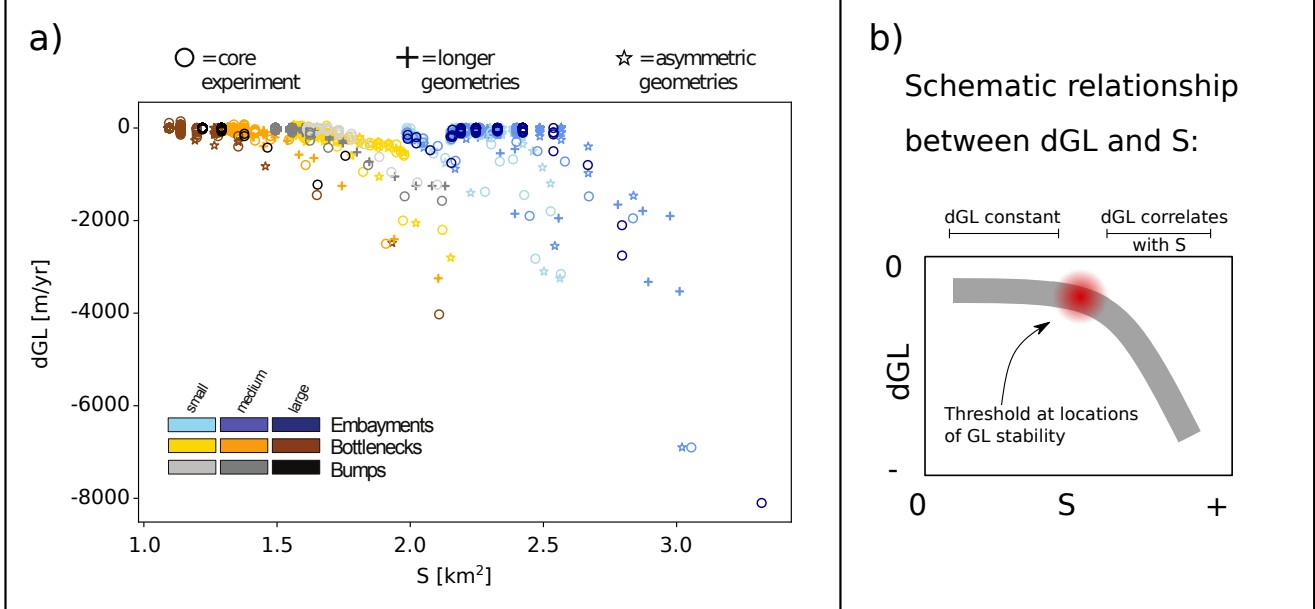

**Figure 7.** Relationship between grounding line retreat rate $dGL$ and wetted area $S$. a): All instances when the grounding line is within a geometric perturbation for all tested geometries, except for depressions since retreat in these perturbations is governed by different dynamics than in the ones shown. b): Schematic of a typical relationship between $dGL$ and $S$ where $dGL$ is constant for low $S$, while high $S$ induces faster retreat. The transition between these two states occurs if the GL retreats past a point of intermittent stability. The location of these points may be controlled by either low $S$ or high $dS$.

medium-sized outlet glaciers found in, e.g. Greenland (Carr et al., 2014; Bunce et al., 2018; Catania et al., 2018), Svalbard (Schuler et al., 2020) and Novaya Zemlya (Hill et al., 2017), we expect an even stronger imprint of topography on retreat dynamics.

## 4 Discussion

### 4.1 Mechanisms behind geometric controls of glacier dynamics

The current study offers new quantitative insights into how topography influences the evolution of marine outlet glaciers, and their response to ocean warming. We demonstrate that two topographic metrics, the wetted area $S$ and its derivative $dS$, jointly control the dynamics and retreat of glaciers constrained by fjord walls. Together, these metrics largely explain variations in grounding line mass flux $Q_{GL}$, which is important in the context of sea-level rise, and the grounding line retreat rate $dGL$.

Based on our stress analysis and physical principles, we propose the following physical interpretation for these results: First, a downstream narrowing or shoaling fjord (positive $dS$) stabilizes the glacier, as ice flow is funneled through the constriction enhancing the glacier-fjord contact (Fig. 5a,c). This increases the basal or lateral resistance to flow, which stabilizes the glacier.





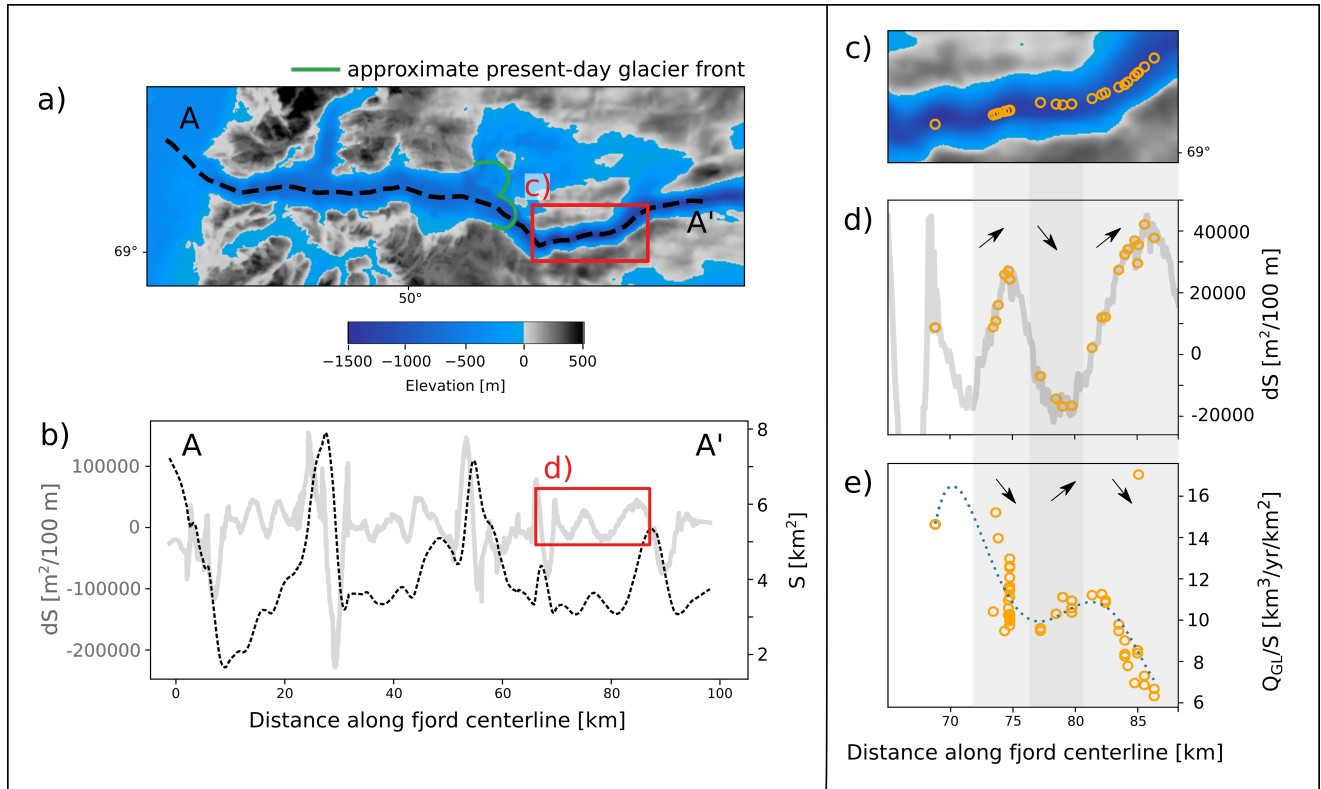

**Figure 8.** The real-world example Jakobshavn Isbræ. a) Topography of Jakobshavn Isfjord (Morlighem et al., 2017) with approximate present-day glacier front, and centerline along which profiles shown in b) of the wetted area $S$ and its along-fjord change $dS$ are calculated; c) zoom to area where JI is enclosed between fjord walls, with yellow circles showing all modeled grounding line positions in this section (Kajanto et al., 2020); d) $dS$ profile in the same section of the fjord with grounding line positions indicated; e) $Q_{GL}/S$ in this section with grounding line positions and a polynomial fit (blue dotted line). The opposing trends in $dS$ (d) and $Q_{GL}/S$ (e) as indicated by the arrows demonstrate qualitatively that the negative relationship $Q_{GL}/S$ over $dS$ can be found in this complex setting.

Conversely, a downstream widening/deepening fjord (negative $dS$) provides little support to the glacier as glacier-fjord contact is reduced (Fig. 5b,d). Second, a narrow fjord (low $S$) stabilizes the glacier, because the distance between the lateral ice margins, where friction with the fjord walls is high, and the center of flow is small. This means that the part of the glacier where ice flow is largely undisturbed is reduced (Raymond, 1996; van der Veen, 2013). Third, a shallow fjord (low $S$) stabilizes the glacier because the glacier is further away from flotation, and thus grounding line retreat is less likely to occur with a given amount of thinning (Pfeffer, 2007; Enderlin et al., 2013). In our experiments, the area exposed to ocean melt does not have a large effect on retreat dynamics. Even high oceanic melt rates, which could compensate for a small ice-ocean interface, do not trigger retreat through geometric perturbations where $S$ is low.

For a particular fjord geometry, the relative importance of $S$ or $dS$ in providing stability to the glacier may vary. This will be discussed with two examples from our results: In embayments, $S$ is larger than the reference fjord. Therefore, if $S$ was the





dominant control for glacier dynamics here, retreat through embayments should occur more easily than through the reference fjord. However, we find that the opposite is true; the glacier stabilizes at the downstream end of embayments (Fig. 3a), while

it retreats steadily through the reference fjord (not shown). This indeed confirms that $S$ alone does not explain glacier retreat. Rather, $dS$ controls glacier dynamics, because the point of intermittent grounding line stability is where the fjord changes from wide to narrow in the direction of ice flow. For bed bumps, the picture is different. Our model glaciers stabilize on or near the crest of the bumps, where $dS$ is close to 0 or negative (Fig. 3d). This should not be an obstacle for retreat if $dS$ was the dominant control on glacier dynamics. Therefore, it must be the shallowness of the fjord at this point (indicated by low $S$) which governs

the dynamics here.

    Given these disparities between different settings, it is all the more compelling that we find the geometric relationship $Q_{GL}/S$ over $dS$ universal to all our tested fjords. It implies that given the current grounding line mass flux $Q_{GL}$ and the upstream subglacial topography of a particular glacier, a well-founded estimate of the topographically-induced future contribution to sea-level rise can be made. To the authors knowledge, this type of universal quantitative link between fjord topography

and glacier response has not been established before, going beyond the qualitative descriptions of ice-topography interaction offered in previous studies (Enderlin et al., 2013; Carr et al., 2014; Bunce et al., 2018; Catania et al., 2018; Åkesson et al., 2018b). For projections of future sea-level rise, this direct coupling between topography and ice discharge is highly relevant, as it enables an ad-hoc assessment of the expected future sea-level contribution of a glacier on decadal to centennial time scales. $Q_{GL}$ is readily available for glaciers where the velocity and bathymetry is well-known. Moreover, where bathymetry is

uncertain, or for less well-studied glaciers, we demonstrate that the velocity evolution over time is also a good proxy for the dynamic reaction of a glacier to fjord topography (Fig. SC1).

    Our second quantitative relationship between $dGL$ and $S$ confirms the widely accepted concept that a wide or deep fjord promotes fast grounding line retreat (e.g. Warren and Glasser, 1992; Enderlin et al., 2013; Carr et al., 2013; Bunce et al., 2018; Catania et al., 2018; Åkesson et al., 2018b). However, we highlight that this relationship may not hold if the fjord is narrowing

or shoaling downstream (positive $dS$). In practical terms, this means that retreat from a uniformly wide and deep channel into an upstream widening or deepening section does not automatically imply that retreat has to accelerate. Rather, we suggest that a glacier will sit at the downstream end of the section, where $S$ is increasing upstream, for a considerable time, or may not even retreat further, because it is particularly stable here. Only if this position is abandoned, fast retreat will occur (Fig. 5a). This fast retreat is in fact facilitated by the long residence time of the grounding line, since concurrent upstream thinning preconditions

the glacier for fast retreat.

    In contrast to most previous studies, we emphasize the role of along-fjord change in fjord topography ($dS$) to explain geometric controls of outlet glaciers. Along-fjord change in fjord depth is also key in the context of the marine ice sheet instability (MISI) theory, according to which retrograde beds promote retreat (Schoof, 2007; Gudmundsson et al., 2012; Gudmundsson, 2013). However, even though we do have retrograde beds in our fjords with basal perturbations, we do not see any influence

of the MISI on glacier dynamics. This is simply because our tested glaciers never retreat into an area where the bed slope is strongly negative, and where the MISI effect would be expected to occur. On bumps, the glaciers stop to retreat on the downstream side where the bed is prograde (Fig. 3d). In depressions, the grounding line is intermittently stable where the bed




slope is only slightly negative (Fig. 3c), which is not enough to trigger a MISI feedback loop. Retreat off this stable position
occurs through ungrounding several kilometers upstream of the grounding line. This process is not related to typical MISI dy-
namics. Besides that, our quantitative relation $Q_{GL}/S$ over $dS$ may seem contradictory to widely accepted concepts of glacier
dynamics, because we project high $Q_{GL}/S$ for prograde beds (i.e. negative $dS$ in our study). This may give the impression
that prograde beds should lead to accelerating ice discharge. However, we emphasize that we assess ice discharge *per area*, not
absolute values of ice discharge. A glacier retreating on a prograde bed will experience a reducing wetted area as it recedes,
and thus the ratio $Q_{GL}/S$ may increase, but not the absolute grounding line flux. This is exemplified by our experiments with
bumps, where the glacier stabilizes on a prograde bed even though $Q_{GL}/S$ is relatively high (Fig. 6). Essentially, we thus
describe a mass conservation mechanism where a smaller flux gate requires a smaller ice flux to maintain the same grounding
line position. This is well known in the context of ice-topography interaction (Jamieson et al., 2012; Åkesson et al., 2018b).

Only few studies have considered the influence of along-fjord changes, rather than absolute values, in fjord width on glacier
dynamics, and available observations are limited in time (Carr et al., 2014; Bunce et al., 2018). The main consensus is that a
fjord widening in the direction of glacier retreat promotes fast grounding line recession, while a narrowing fjord reduces retreat
rates. Furthermore, retreat onto a pinning point can stabilize the grounding line. This is related to our findings in that we also
see accelerating retreat the further the grounding line moves into a wider fjord upstream (c.f. grounding line positions in the
downstream half of the embayment (55 km < x < 65 km) in Fig. 3a). However, in our results, this only occurs after a phase of
grounding line stability at the downstream end of such fjord sections. We do not find conclusive evidence in the observational
record whether these points of intermittent grounding line stability are a relevant phenomenon in real-world settings or not
(Carr et al., 2014; Bunce et al., 2018; Catania et al., 2018). Further research analysing a range of fjord geometries and glacier
retreat histories is required to test this result. We do see some signs in our experiments that retreat slows down the further the
grounding line recedes into a narrower fjord upstream. Overall though, retreat in upstream narrowing fjords is markedly faster
than if the fjord is upstream widening (compare grounding line positions in the downstream half of bottlenecks (55 km < x < 65
km) with the ones in the upstream half (45 km < x < 55 km) in Fig. 3b). This we explain with the aforementioned enhancement
(reduction) in fjord-glacier contact for an upstream widening (upstream narrowing) fjord (Fig. 5a,b). Thus, we confirm that
retreat slows down in an upstream narrowing fjord, but in the context of a retreat cycle through both upstream widening and
narrowing fjord sections, overall faster retreat occurs through upstream narrowing fjords. Therefore, the observational records
of glacier retreat in Greenland may be too short and the fjord-width variations too small to testament similar dynamics as we
observe in the model (Carr et al., 2014; Bunce et al., 2018; Catania et al., 2018). However, in line with the observational record,
we can reproduce the stability that lateral pinning points offer. This is clearly demonstrated by the strong stability that narrow
bottlenecks provide in our experiments.

## 4.2 Study limitations

As in all numerical studies, our results have limitations related to the choice of model parameters. In particular, this concerns
the friction coefficient, which we assume spatially uniform to isolate the signal of the dynamic component of ice loss, regardless
of where the glacier is situated in the fjord. Given that water availability is typically higher at the bed compared to at the fjord





walls, other approaches could be considered in future studies. Setting a friction coefficient increasing with altitude (Åkesson et al., 2018a; Kajanto et al., 2020) for instance, may reduce the potential of basal perturbations to stabilize the glacier relative to lateral perturbations. This may be one reason why $Q_{GL}/S$ values for lateral perturbations were found to be higher than

the ones for basal perturbations, and why no glacier could be enforced to retreat over bumps. Another reason may be that the depth-integrated ice-flow approximation (SSA) overestimates the stabilizing effect of a high basal friction point on the vertical velocity profile compared to a 3D model, thus rendering bumps excessively effective in stabilizing the glacier (MacAyeal, 1989).

The idealized setup also means that potentially complex processes, such as the ocean-induced melt or the subglacial hydrol-
ogy, are parameterized in a simple way. This needs to be considered when applying our results to real-world glaciers, and it means that the geometric relationship found here may not be as distinct in complex settings. This is what we see in our case study of Jakobshavn Isbræ. To what degree our results are transferable to a specific glacier will depend on to what degree the stress regime at the glacier front and the grounding line is comparable to our setup. For instance, any floating ice present during our experiments did not provide a significant buttressing effect (Fig. 5). Therefore, ice - topography interactions in the vicinity
of the grounding line were found important. For settings where an ice shelf significantly alters the stress regime, topographic control of the glacier could arise from processes at the ice-shelf front rather than the grounding line. Overall, our experimental setup is designed to represent typical marine outlet glaciers in Greenland and the Arctic, and values for parameterizations of, for example, the ocean forcing, basal friction, and iceberg calving have been set accordingly. The idealizations concern parameters that modulate a glaciers response to a specific climate forcing as well, but we do not expect them to be a first-order
control on the geometrically induced dynamics. Therefore, we expect our findings to be applicable to a wide range of outlet glaciers.

Many of the medium-sized glacier catchments in Greenland may yield a long-term contribution to sea-level rise as thinning at the outlet glaciers may propagate far upstream (Felikson et al., 2021). Here, we offer a quantitative perspective on the processes at the grounding line, and highlight the importance of assessing both the wetted area and the along-fjord change in wetted area
in order to accurately describe ice-topography interactions. These two parameters together determine the geometrically induced ice discharge to the ocean, which is crucial for sea-level rise, and the expected future retreat of marine outlet glaciers.

## 5 Conclusions

The shape of a fjord can promote or inhibit glacier retreat in response to climate change. Here, we use a numerical model to study such ice-topography interactions in a synthetic setup under idealized conditions. We find that variable fjord topography
induces gradients in lateral or basal shear stresses which then influence glacier dynamics. Increased shear at the ice-fjord interface, which stabilizes the model grounding line, is caused by converging flow towards a downstream constriction, because such flow enhances the glacier-fjord contact. Conversely, areas of reduced shear, which promote fast retreat, are found where ice flow diverges, because glacier-fjord contact is reduced. In practical terms, this means that retreat of a glacier into an upstream widening or deepening fjord does not necessarily promote retreat, but may in fact stabilize the glacier. We also confirm that

rapid retreat is more likely to occur through deep and wide fjords, while slower retreat is expected for narrow and shallow topographies.

Furthermore, using the concept of the wetted area, which is the submerged cross-sectional area of a fjord, and its along-fjord change, we quantitatively link grounding line discharge and retreat rate to fjord topography. Specifically, we postulate that given the current grounding line flux and the upstream subglacial topography of a particular glacier, an ad-hoc estimate of the

topographically-induced component of future mass loss can be made. For less well-studied glaciers, we demonstrate that the velocity evolution over time is a promising proxy for the dynamic response of a glacier to fjord topography.

We expect that the quantitative relationships between topography and retreat dynamics are most likely to be transferable to real-world glaciers confined by fjord walls, while being less relevant for ice streams where lateral ice flow influences dynamics considerably. We demonstrate this using the example of Jakobshavn Isbræ.

Future studies should aim to verify our findings using real-world observations. Particularly valuable in this context would be long-term observations of (sub-)annual grounding line positions, calving fronts and velocity changes, combined with detailed bathymetric maps for glaciers confined by fjord walls in Greenland and the Arctic.

*Code availability.* ISSM 4.16 is freely available for download at http://issm.jpl.nasa.gov/. Key scripts needed to perform the simulations done in this study will be made publicly available on GitHub once the manuscript is published.

*Author contributions.* HÅ, KHN and BdF came up with the idea for the project. TF designed the model setup with significant input from HÅ, BdF and MM. TF carried out and analyzed the experiments with input from all co-authors. TF wrote the manuscript and produced the figures with feedback from all co-authors.

*Competing interests.* The authors declare that they have no known competing financial interests or personal relationships that could have appeared to influence the work reported in this paper.

*Acknowledgements.* HÅ was supported by the Swedish Research Council, grant no. 2016-04021. BdF is funded by the Norwegian Research Council (grant 287206). The simulations were performed on resources provided by UNINETT Sigma2 - the National Infrastructure for High Performance Computing and Data Storage in Norway (nn9635k, nn4659k). We thank Karita Kajanto for providing model output on Jakobshavn Isbræ used in this study.

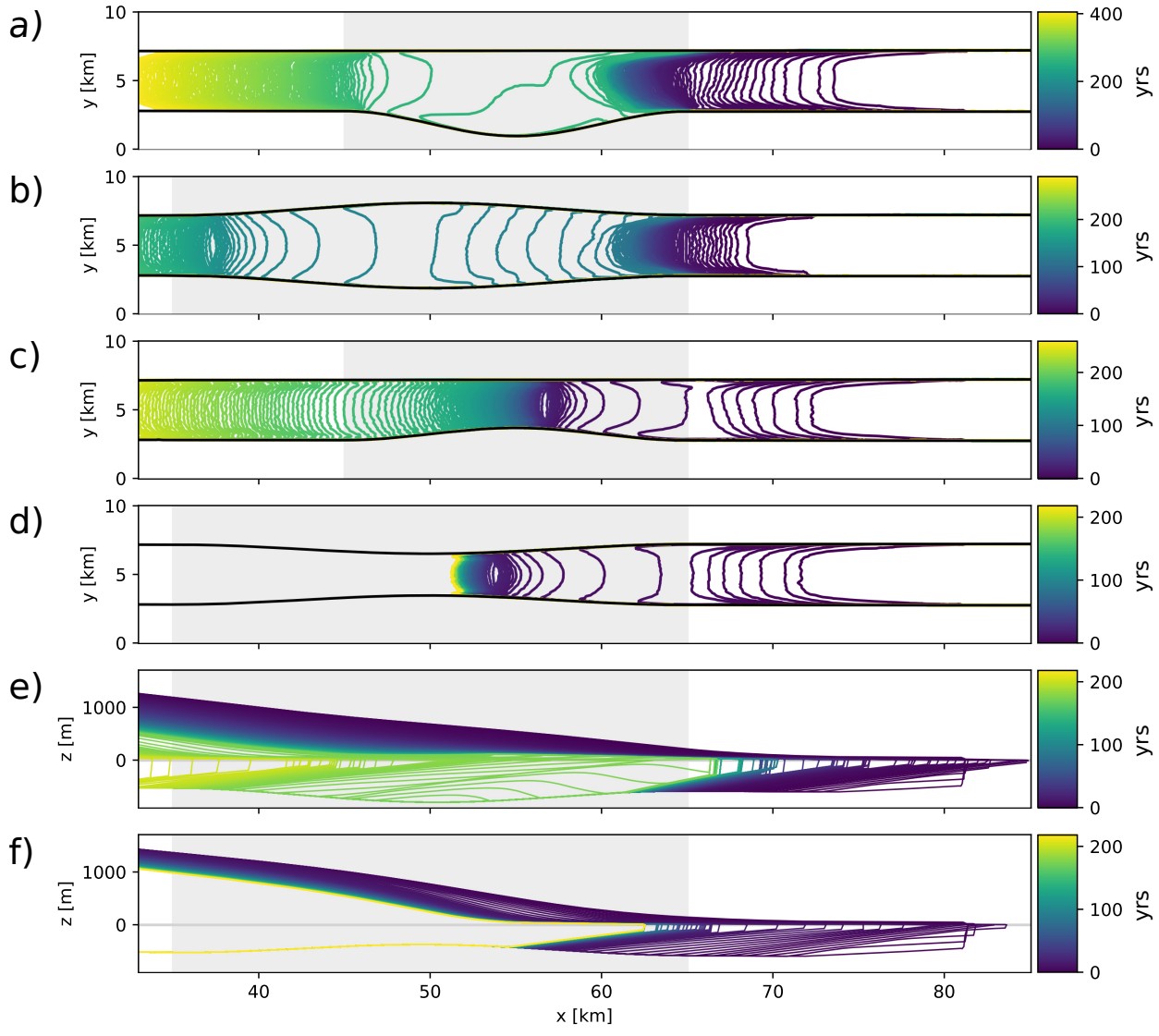

**Figure A1.** Retreat through *asymmetric* and *longer* perturbations. Annual grounding lines for lateral perturbations (a-d), annual profiles through glaciers for basal perturbations (e,f). Geometries shown are (with names referring to Table 2): a) ByH1800_asy, b) ByH900_lon, c) ByH-900_asy, d) ByH-675_lon, e) BuH-240_lon, f) BuH180_lon. Shaded areas indicate extent of geometric perturbation.

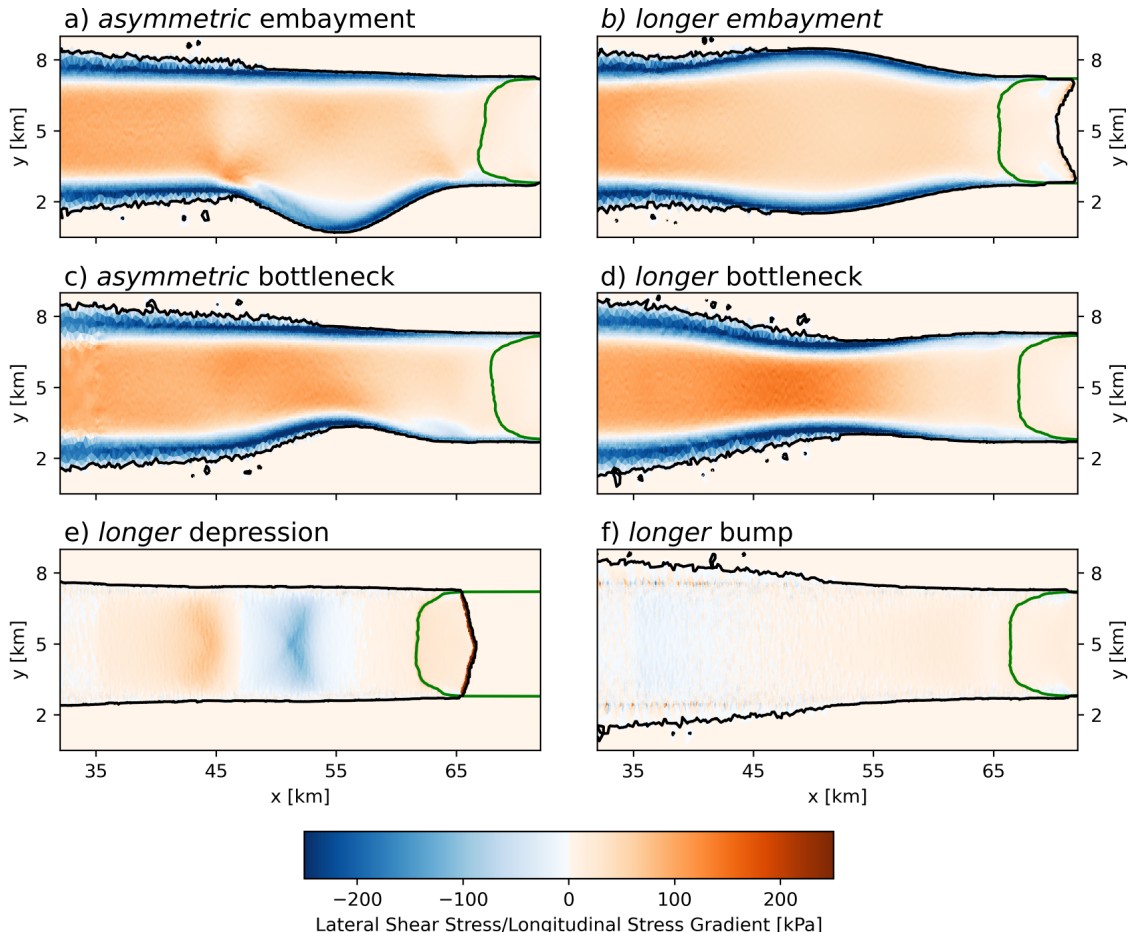

**Figure B1.** Stress fields in *asymmetric* and *longer* geometries. Lateral shear stress for lateral perturbations (a,b,c,d) and longitudinal stress gradients for basal perturbations (e,f)





**Figure C1.** $v_{GL}$ over $dS$ correlation for all geometries including asymmetric and longer ones. Differences between this relationship and the relationship $Q_{GL}/S$ over $dS$ originate from time instances when the ice front is grounded well above flotation (and therefore $v_{GL} \not\propto Q_{GL}/S$) and because we measure $v_{GL}$ only along the central flow-line of the glacier.



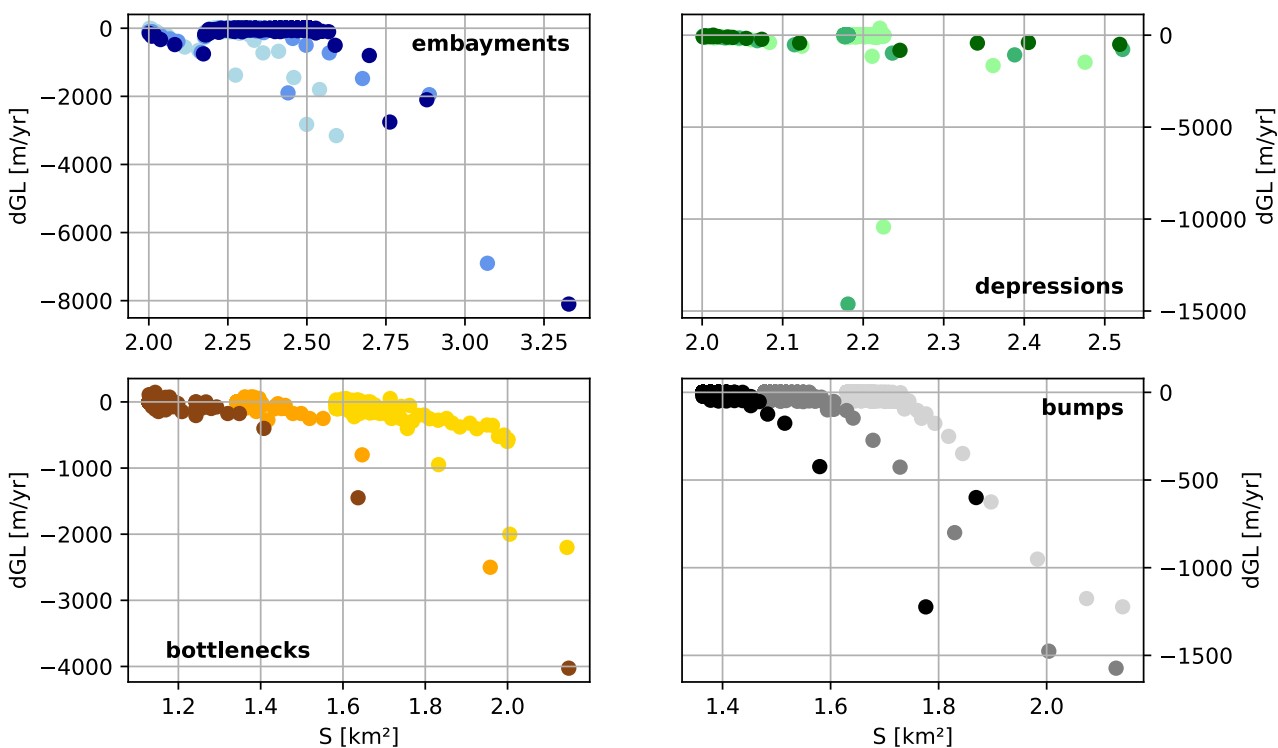

**Figure D1.** *dGL* over *S* correlation including depressions.

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
