# Peer review of "Geometric Controls of Tidewater Glacier Dynamics"

_The Cryosphere, 2021_

## Author Comment (AC2)

Supplementary figures for response to reviewer #2 concerning the manuscript "Geometric controls of Tidewater Glacier Dynamics" by Frank et al., (2021), TCD

Frank, T., Åkesson, H., de Fleurian, B., Morlighem, M., Nisancioglu, K.H.

July 2021

Figure S1: Grounding line position over time for different fjord geometries and magnitudes (for color code refer to Fig. 2 in the manuscript). From top to bottom: linear-fjord reference run, embayments, depressions, bottlenecks, bumps

Figure S2: Relationship between velocity at the grounding line and the wetted area S for all geometries tested. For color code, please refer to Fig. 6 in the manuscript.

---

## Author Response (AR1)

**Geometric Controls of Fjord Glacier Dynamics - response to reviewers**

Frank, T., Åkesson, H., de Fleurian, B., Morlighem, M., Nisancioglu, K.H.

July 2021

**1 Response to Anonymous Referee #1**

We would like to thank the reviewer for taking the time to review our manuscript, and we appreciate the positive attitude towards it as well as the helpful comments to further improve it. We reply below by highlighting the reviewer's comments in italics above our answers. Changes made to the text are indicated under each of our answers in blue.

*My main comment is the discussion of bumps and no or slow retreat of the GL while the authors use SSA - plus no subglacial hydrology, and the Budd type friction law with a uniform friction coefficient. I understand the choices for the experiment setup but would wish for a more detailed discussion how a higher order or full stokes simulation would affect the results on bumps and overdeepenings (e.g. SSA critised for a similar setup in Favier et al. (2012)). Furthermore, this concern is only raised in sub-section 4.2. Before (Lines 217ff, 263, 367ff, ...), it is only mentioned that the reason for no retreat at bumps is the shallowness of the fjord.*

We acknowledge that the SSA is not a full representation of the complex stress regime near the grounding line. For weak beds and fast-flowing outlet glaciers, as we aim to mimic in our synthetic setup, the SSA is however a reasonable approximation and widely used in the glaciological literature. In the study cited above (Favier et al., 2012), we do not see any criticism for using the SSA in a similar setup. The comparison of their results using a full-Stokes model to a setup using the SSA refers to a study by Goldberg et al. (2009), but, as is stated in Favier et al. (2012): "The authors [Goldberg et al. (2009)] also observed a merge between both grounded areas with advance rates of the grounding line situated around 70 m a$^{-1}$, whereas in our case the rates are lower and between 15 and 30 m a$^{-1}$. This discrepancy is not surprising since both experiments were performed under different configurations and are hence not directly comparable. For example, the SSA experiments were conducted with a null velocity prescribed along the lateral walls, whereas we imposed free slip." (Favier et al., 2012). We do not see how this statement, or any other made in Favier et al. (2012), discount the appropriateness of using the SSA in a setup similar to ours. On the contrary, we find that the results presented there using a full-Stokes model agree favorably well with our SSA results for bumps and overdeepenings. In Fig. 11 of Favier et al. (2012), longitudinal deviatoric stresses are plotted alongside along-flow profiles through the glacier. The stress distributions show the same characteristics as in our results (Fig. 5), with compression on the stoss side of a bed rise, and extension on the lee side. Furthermore, the along-flow profile show that on top of the pinning point in Favier et al. (2012), the glacier surface shows a bulge which is exactly the same feature that we observe in our setup (Fig. 3c.). Therefore, we believe that the study by Favier et al. (2012) rather supports our methodology in that it shows that

a full-Stokes or higher-order model would likely yield similar results as the setup with SSA used here, and that the conclusions would remain.

With this said, we will expand on the discussion on the influence of the ice-flow approximation, following the discussion above. This will be done in 4.2 Study limitations, but also at appropriate places in the manuscript (e.g. line 217ff, 376ff, 435ff).

Added in line 82f: A discussion on the appropriateness of this approximation [i.e. using the SSA] compared to a Full-Stokes model is provided in sect. 4.2

Added in line 107ff: The effective pressure in the friction law induces an elevation-dependence of the basal friction. This dependence is motivated through the assumption that weak sediments are more likely to be present in low-lying areas. Implications for our results are discussed in sect. 4.2.

Rewrote section 4.2 (line 447ff): In particular, there are three aspects that warrant further discussion. First, the SSA used here is not a full representation of the stress regime in a glacier which especially bears relevance near the grounding line. For weak beds and fast-flowing outlet glaciers, as we aim to mimic in our synthetic setup, the SSA is a reasonable approximation and widely used in the glaciological literature. It may fail, however, on steep bed slopes, such as the ones in our simulations of basal perturbations. In particular, the effect of basal high-friction points may be overestimated which might explain why basal perturbations generally yield lower $Q_{GL}/S$ values than lateral perturbations in our experiments (Fig. 6), and why bumps were found to be excessively stabilizing. However, we do not expect this effect to compromise our conclusions in general. For instance, Favier et al. (2012), using a setup similar to ours to simulate the effect of a basal pinning point on ice dynamics with a Full-Stokes model, present stress distribution patterns that agree favorably with the ones shown here. The influence of using the SSA as compared to a Full-Stokes model on our results is therefore expected to be limited. Second, choosing a Budd-type friction law which introduces an elevation-dependence of the basal friction through the effective pressure adds complexity to the interpretation of our results. Specifically, it means that bed bumps and depressions alter the basal resistance to flow purely through their elevation difference to the surroundings. This may be another reason why no retreat over bumps was possible in our experiments. Furthermore, while the Budd friction law is one of the most commonly used ones, previous studies have also shown that different friction laws can lead to substantial differences in transient ice dynamics and steady-states (Brondex et al., 2017; Åkesson et al., 2021). A comparison between different friction laws is outside the scope of this study, and thus the impact of this modeling choice is difficult to estimate. Third, the calving parameterization chosen is an important control on the simulated dynamics. Other idealized studies have applied a calving law with a prescribed ice front position or a prescribed ice shelf length (Schoof et al., 2017; Haseloff and Sergienko, 2018), which both have the disadvantage of being unknown in general, whereas we opt for the von-Mises parameterization due to its relatively good performance when applied to real-world glaciers (Choi et al., 2018). Again, in the absence of a universal calving law, the potential effect of this modeling choice on our results is hard to assess. Overall, it can be assumed that the geometric relationship found here is not as distinct in complex real-world settings. To what degree our results are transferable to a specific glacier will depend on the degree to which the stress regime at the glacier front and the grounding line is comparable to our setup. This is what we demonstrated with the example of Jakobshavn Isbræ. Nonetheless, since we use a state-of-the-art model with parameterizations that have been calibrated against typical values for marine outlet glaciers, we expect our findings to be applicable to a wide range of settings.

*The special section 4.2 "Study limitations" makes sense in this manuscript and addresses the issue (only briefly) but should not be disconnected from the results and discussion. I highly recommend referring to section 4.2 throughout the text and/or change the strong conclusions (like L.369 "Therefore, it must be the shallowness of the fjord at this point (indicated by low S) which governs the dynamics*

*here.") to more weaker expressions and elaborate a bit more on line 435ff.*

Good point; we will make sure that the assumptions and limitations are more clearly articulated throughout the manuscript. Note, however, that the quote above is placed in context of comparing results of our artificial fjord settings where dS is the dominating control on retreat against settings where S is the governing factor. With this specific phrasing we therefore do not intend to draw any general conclusions about real-world glaciers, which would require a more detailed discussion on limitations, at this specific point in the text. We will clarify this distinction in a revised manuscript. Regarding line 435ff, we suggest to add a short explanation that not resolving vertical shear is less accurate where the bed slope is relatively steep, but that we do not expect our results to be hugely influenced as is pointed out in the comparison with a full-Stokes simulation above.

Besides additions to the text mentioned above: Added in line 98ff: Note that the choice of the calving law is an important, yet poorly constrained, control on the grounding line dynamics and ice front behaviour (Schoof et al., 2017; Haseloff and Sergienko, 2018). In the absence of a universal calving law, a reasonable assumption has to be made, and we justify the choice of the von-Mises law through its relatively good performance in real-world applications (Choi et al., 2018).
Added in line 273f: However, the strong stability that bumps provide to the glacier may also be related to the choice of model parameters (sect. 4.2.).
Added in line 316: (note that this may be influenced by our modelling choices (sect. 4.2))
Added in line 379: in our experiments

*I assume you don't see any advance during your simulations since you set-up the experiment to grounding line retreat. However, I suggest including a comment (in section 4), how and if your finding could also capture/predict GL advance? If your relationship of GL retreat and wetted area (or velocity) is to be applied in real-life, those areas might also have seen GL advance.*

These are good suggestions that we are happy to address in the revised manuscript. Indeed, we do not see any grounding line advance, but previous studies (Åkesson et al., 2018; Brinkerhoff et al., 2017) have shown that fjord geometry induces hysteresis in the retreat-advance cycle, where readvance is inhibited if a fjord is widening or deepening ahead of a glacier. We expect this to hold for our experiments as well.

Added in line 440ff: Our experiments are set-up to simulate grounding line retreat, and hence we do not offer any insights on ice-topography interactions for advancing glaciers. Previous studies, however, suggest that fjord geometry induces hysteresis in the retreat-advance cycle of a glacier, meaning that a reversal to colder conditions after a phase of climate warming does not allow the grounding line to advance to the same position it occupied initially if the fjord is widening or deepening in front of the glacier (Brinkerhoff et al., 2017; Åkesson et al., 2018). We expect this also to hold for our experiments if we had simulated ocean cooling following the warming scenarios tested.

**1.1 Specific comments**

*- L.6 Re-phrase, it is a bit confusing for an abstract: "We find that retreat in an upstream widening or deepening fjord does not necessarily promote retreat, but conversely, [...]"*
Rephrased as: "We find that retreat in an upstream widening or deepening fjord does not necessarily promote retreat, as suggested by previous studies. Conversely, ..."

*- L.65 "larger suit of experiments" – why not mention you do 21 simulations?*
Rephrase as: "Here, we use a numerical ice-flow model resolving two horizontal dimensions, we

include a suite of 21 experiments and present a systematic approach..."

*- L.143-154: The table 2 is very clear, but the text is partly confusing. For example: "we test 20 fjord geometries [...] Additionally, we test [...]" Clarify this paragraph in term of total numbers and subsets.*

changed "additionally" to: "In the eight simulations outside our core experiments, we test asymmetric and longer perturbations to verify..."

*- L.206 Define "very slowly" and "retreats quickly"*

We define slowly as $> -100$ m $yr^{-1}$ and quickly as $< -500$ m $yr^{-1}$.

L212 now reads: We identify positions of GL stagnation ('stagnant' GL positions), i.e. where the GL rests for a sustained time (typically 50 to 200 yrs), or retreats slowly ($dGL > -100$ m yr$^{-1}$), and areas where the GL retreats quickly ('ephemeral' GL positions; $dGL < -500$ m yr$^{-1}$).

*- L.226: Please re-phrase for clarity: "However, in fjords that have a smaller S at $x_C$ than the reference fjord (bottlenecks and bumps), stable positions are also found where the fjord is narrow or shallow (small S). Therefore, S is also an important control on GL retreat."*

Rephrase as: However, glaciers in narrower or shallower fjords than the reference fjord (bottlenecks and bumps) can also temporarily stabilize where S is small. This shows that the wetted area constitutes an additional important control on GL retreat.

*- Fig. 8a: Maybe adjust present day glacier front line colour...*
Done.

*- Fig. A1e&f: include line for bed topography (like in Fig. 3 c,d)*
Done.

**2 Response to Anonymous Referee #2**

We thank the reviewer for the thorough assessment of our manuscript! We appreciate the constructive comments as they highlight where we need to justify our modelling choices more rigorously. We will address the comments as outlined below.

**2.1 Methods**

*Why include an elevation-dependence for the flux at the inflow boundary and why have accumulation only in a small area upstream rather than a constant accumulation rate on the entire geometry as is typical in idealised modelling studies? Including the elevation dependence adds an unnecessary complexity that does not provide any additional insights.*

The elevation dependence at the influx boundary is a way to parameterize the surface mass balance - altitude feedback. We acknowledge that including this feedback is not strictly necessary for the idealized experiments conducted here. However, we believe there are some good reasons for doing so: First, this feedback is well-established in glaciological theory (e.g. Harrison et al., 2001) and has been shown to play a vital role in glacier and ice-sheet evolution (e.g. Åkesson et al., 2017; Boers and Rypdal, 2021). We believe that not including it is a clear misrepresentation of reality and that the realism gained by including it outweigh the slight loss of simplicity. It is correct that we have simplified many of the complex processes governing glacier dynamics to reduce the number of degrees of freedom for the interpretation, but this is because many of these processes are poorly understood,

badly resolved in models or costly to simulate. For the surface mass balance - altitude feedback, this is not the case. Second, the variation in influx is overall small compared to the total mass gains of the glacier. As is given by eq. 3 in the manuscript, the mass gain through surface accumulation amounts to $5.5 \times 10^9$ m$^3$ yr$^{-1}$, which can be compared to a mass gain of $1 \times 10^9$ m$^3$ yr$^{-1}$ through influx with a thickness of about 2 km at the influx boundary for our steady-state glacier. Towards the end of our simulation, the thickness at the influx boundary will have reduced depending on the fjord geometry, but not by more than ∼1400 m (corresponding to an ice thickness of 600 m at the influx boundary). This minimum thickness translates to an influx of $0.3 \times 10^9$ m$^3$ yr$^{-1}$ which implies a reduction in total mass gains of only about 11% over the entire simulation. So even under the assumption that including the parameterized surface mass balance - altitude feedback is unnecessary, it does not have a dominating influence on the retreat dynamics. Finally, it is worth pointing out that setting a constant accumulation rate on the entire geometry, as suggested by the reviewer, would also have led to a reduction in mass gain in the course of glacier retreat because the surface area of the glacier decreases as it recedes.

no changes to the text

*Why use the Budd-sliding law? This sliding law includes the dependence on bed elevation below sea level, which introduces additional complexity that can obscure the results (see also Brondex et al., 2017, for a study of the sensitivity of grounding line dynamics to the choice of the friction law). Where there are bedrock bumps or dips, this changes the basal resistance to flow.*

The type of friction law is indeed one of the many model choices that inevitably will influence the results. Without an explicit subglacial hydrology model, basal drag needs to be parameterized. Many studies, including the current one, assume some dependence on bed elevation, either through the effective pressure (e.g. Morlighem et al., 2019; Åkesson et al., 2021), or through an elevation-dependent basal friction parameter (e.g. Aschwanden et al., 2019). This is true for Budd-type laws (where the basal effective pressure is given as $N = \rho_i g H - \rho_w g \max(0, -z_B)$ (cf. eq. 2), as well as for some Coulomb-type laws. There is some physical rationale behind this elevation-dependency, whether implied through the effective pressure (as in the current study) and/or subsumed into a friction parameter (e.g. Aschwanden et al., 2019; Åkesson et al., 2021); ice generally flows faster further downstream (low elevations) than upstream (high elevations), and potentially weak sediments are more likely to be present in low-elevation areas. Again this will be a trade-off between real-world aptness and idealized simplicity.

The Budd law is one of the most widely used in the literature and we therefore think that our findings will be relevant to the wider glaciological community. A comparison across different friction laws is beyond the scope of the current study, and has indeed been done before in a slightly different setting, as pointed out by the reviewer. Nevertheless, we will follow the reviewer and include a more thorough discussion on the choice of friction law and the potential biases introduced. Specifically, we will clarify that the Budd law introduces an elevation-dependency, through the parameterization of the effective pressure, that is particularly critical for the fjords with bumps.

Added in line 107ff: The effective pressure in the friction law induces an elevation-dependence of the basal resistance to flow. This dependence is motivated through the assumption that weak sediments are more likely to be present in low-lying areas. Implications for our results are discussed in sect. 4.2.

Added in line 457ff: Second, choosing a Budd-type friction law which introduces an elevation-dependence of the basal friction through the effective pressure adds complexity to the interpretation of our results. Specifically, it means that bed bumps and depressions alter the basal resistance to flow purely through their elevation difference to the surroundings. This may be another reason why no

retreat over bumps was possible in our experiments. Furthermore, while the Budd friction law is one of the most commonly used ones, previous studies have also shown that different friction laws can lead to substantial differences in transient ice dynamics and steady-states (Brondex et al., 2017; Åkesson et al., 2021). A comparison between different friction laws is outside the scope of this study, and thus the impact of this modeling choice is difficult to estimate

*Use of different forcings to trigger retreat: again, this makes it difficult to compare different results. Why not use one (strong) melt forcing for all cases?*

As is described in Section 3.2, we see the magnitude of forcing to trigger full retreat as a source of information, rather than a hinder to compare the results. This is because the forcing strength is one of two metrics to compare how efficient the different fjord geometries are in influencing glacier retreat (the second one being the residence time of the grounding line in a position of intermittent stability). For instance, had we simply chosen one strong melt forcing, we would most likely not be able to determine which of the small depression, medium depression or large embayment provide most stability, as the glaciers in these fjords all retreat after about the same residence time (Fig. 4). Furthermore, to choose the same strong melt forcing for all geometries would probably have induced faster retreat dynamics in those fjords where the glaciers also retreat with a smaller forcing. This would have reduced the level of detail and hampered our insight into the retreat dynamics. Finally, real tidewater glaciers react quite promptly to changes in ocean forcing (Khazendar et al., 2019). If, for instance, a slightly strengthened ocean forcing triggers glacier retreat, this will unfold immediately and not be delayed as the forcing strengthens further. For a study like ours, where we want to mirror this behaviour, this implies that it is the most realistic approach to assess glacier response to a forcing that is as small as possible. Therefore, we have chosen to try multiples of the reference forcing to trigger full retreat, and go with the smallest possible value. In our view, this is more realistic than setting one very strong value for all glaciers, because a real glacier would have reacted to a warmer ocean long before that ocean has warmed by a very large amount.

no changes to the text

*Lines 160-163 mentioning that doubling the melt rate leads to a reduction in calving and the grounding line is mostly stable. Again, such a choice of calving law is unfortunate, as it is not clear which dynamics are due to the topographic controls and which are due to feedbacks between melting and calving (see also Schoof et al.; 2017 and Haseloff & Sergienko; 2018 for discussions about how the choice of the calving law can alter grounding line dynamics).*

We acknowledge that the choice of the calving law is an important control on the grounding line dynamics and calving front behavior. As discussed in the studies mentioned above (Schoof et al., 2017; Haseloff and Sergienko, 2018), a calving law using a prescribed ice front position or a prescribed ice shelf length may have produced different results. However, both of these rather idealized approaches seem unsuitable as we have no such information for the future evolution of outlet glaciers. In fact, a comparison between different calving laws for Greenland outlet glaciers has been done before (Choi et al., 2018). One of the laws tested was a height-above-buoyancy criterion which equals the calving-at-flotation criterion discussed in Schoof et al. (2017) if the tuning parameter $q$ is set to 0. However, it was found that this law is not equally well able to reproduce observed retreat patterns as the von-Mises calving chosen in this study. We thus rely on those results in that we choose the calving law that was found most appropriate. In the absence of a universal calving law and considering the scope of our study, we find this approach to be the best available option, even though we are aware of the limitation that comes with it. Note that the calving stress threshold ($\sigma_{max}$ in eq. 1 in the manuscript) has been

set in accordance with typical values for Greenland outlet glaciers, as is explained in the manuscript (line 95). In the revised manuscript, we will more clearly stress the dependence of our results on the chosen calving law.

Added in line 98ff: Note that the choice of the calving law is an important, yet poorly constrained, control on the grounding line dynamics and ice front behaviour (Schoof et al., 2017; Haseloff and Sergienko, 2018). In the absence of a universal calving law, a reasonable assumption has to be made, and we justify the choice of the von-Mises law through its relatively good performance in real-world applications (Choi et al., 2018).

Added in line 464ff: Third, the calving parameterization chosen is an important control on the simulated dynamics. Other idealized studies have applied a calving law with a prescribed ice front position or a prescribed ice shelf length (Schoof et al., 2017; Haseloff and Sergienko, 2018), which both have the disadvantage of being unknown in general, whereas we opt for the von-Mises parameterization due to its relatively good performance when applied to real-world glaciers (Choi et al., 2018). Again, in the absence of a universal calving law, the potential effect of this modeling choice on our results is hard to assess.

*Use of wetted area: I've never come across this term before and have difficulties subscribing to its usefulness (its application in figures 6 & 7 is highly doubtful, see below). What is wrong with just using the amplitude of the perturbation?*

Indeed, to our knowledge, this term is not common-place in the glaciological context, although it is used by Catania et al. (2018) as 'submarine area' (see their Fig. 4c). If it is deemed beneficiary by the reviewer or the editor, we are willing to change the wording from 'wetted area' to 'submarine area'.

Our wording was adopted from civil engineering and hydrology, where the wetted perimeter and the wetted area are commonly used to describe the size of the intersection between water and another media, for example the contact area between water and the hull of a ship (De Marco et al., 2017). Generally, the main advantage of using such a parameter (whether called wetted area or submarine area) as a metric for fjord geometry is that it integrates information about both depth and width of a fjord into one parameter. Our aim here is to find a universal relationship between fjord geometry and glacier response to external forcing; to that end, it is unavoidable to define such a quantity (we will answer below why we do think that its application in Fig. 6 and 7 is meaningful). Even when we designed the fjord geometries themselves, we explicitly wanted to introduce geometric perturbations *of similar magnitude*, but different type. Without the wetted area it would not be possible to define what *a similar magnitude* is when comparing basal perturbations with lateral perturbations. Only with the wetted area, there is a transparent quantitative link between, for example, a medium-sized bump and a medium-sized bottleneck. For real-world glaciers, the wetted area is also useful because it is not always possible to distinguish between basal and lateral perturbation in fjords with complex topography. Meanwhile, the wetted area can always be measured, provided sufficient information on fjord bathymetry is available.

Compared to taking the entire cross-sectional area (a 'flux gate') of the fjord - glacier contact (the wetted area plus the area where the ice is above the water line), the wetted area has the advantage that it does not require information on frontal thickness. This implies that the wetted area can be calculated at any point in time and space for any fjord, regardless of whether a glacier is present at the moment or not. Clearly this shows the convenience of using the wetted area for paleo, present and future studies alike.

no changes to the text

**2.2 Results**

*Line 203 onwards: The terms "stable" and "unstable" refer to steady states, not transients, and are incorrect in this context as steady states are only attained at the beginning and the end of the simulation (see e.g., Strogatz, 2018). This needs to be rewritten to use appropriate terminology.*

We follow the reviewer here and propose to use the term ephemeral grounding line position for what was previously called unstable grounding line position, and stagnant grounding line position for what was before referred to as stable grounding line position. This will be changed in the revised manuscript.

changed stable to stagnant and unstable to ephemeral throughout the manuscript

*The presentation of retreat-results in figure 3 is not ideal as it is difficult to identify the important information from this plot. Better plot grounding line position in the center of the geometry vs. time and include both the results for the reference plot with forcing and the small, medium, and large perturbations. This should make it much easier to see where the retreat is fast and where it is slow and how patterns change with different topographic perturbations.*

If we interpret the comment correctly, the reviewer would like to see a plot like Fig. S1, instead of Fig. 3 in the manuscript. Indeed, this has the advantage that several magnitudes of one perturbation type can be plotted in one subfigure. However, we prefer Fig. 3 as it is, for the following reasons: First, it demonstrates to the reader that we used a 2D model, as opposed to previous studies using 1D flow-line models only (Åkesson et al., 2018, e.g.). This underscores one of the novelties of our study. Second, we think that the current Fig. 3 allows a more intuitive interpretation of the retreat dynamics. At this point in the study, the reader has only seen a sketch of our study design (Fig. 1), and a visualization of the variations of the wetted area and its derivative (Fig. 2). With the figure as proposed by the reviewer, the reader cannot easily draw spatial correlations between retreat dynamics and specific features in the fjord. For instance, we think that it would be difficult to understand that grounding line retreat in embayments slows down where the fjord narrows in the downstream direction, because it is not directly visible from Fig. S1 at what location the fjord is narrowing. Therefore, it would be harder for the reader to gain an intuitive understanding of our results. Finally, Fig. 3 in the manuscript has the advantage that it shows processes which are very important for the retreat dynamics, and which can not be depicted by just plotting the grounding line position against time. For example, this refers to the ungrounding in the central part of depressions, a vital part of how retreat is 'revived' after a period of grounding line still-stand. In summary, we do see the benefit of plotting a figure as suggested by the reviewer, but not at the expense of the current Fig. 3. Instead we suggest to add this as a supplemental figure in the Appendix. Furthermore, we will clarify in the caption of Fig. 3 that the spacing between the lines shows the retreat velocity, with lines that are closer together showing a slower retreat.

Added requested figure in supplement and adjusted text in line 214f: For a comparison of the GL retreat dynamics of all simulations within our core experiment, the reader is referred to Fig. SA1.

*Some of the transient results are interesting and maybe counter-intuitive, but the presentation of the results and the unnecessarily complicated model assumptions make it difficult to trust those to be robust.*

We are aware that our results may, to a certain degree, be influenced by the choice of modelling parameters, as is every modelling study. Besides choosing representative values where needed and applying commonly used parameterizations, we therefore dedicated Section 4.2 to how these choices may influence our results. This section will be extended in a revised manuscript, and references to this section will be inserted at appropriate locations in the text, following the comments by both reviewers.

[Figure]

Figure S1: Grounding line position over time for different fjord geometries and magnitudes (for color code refer to Fig. 2 in the manuscript). From top to bottom: linear-fjord reference run, embayments, depressions, bottlenecks, bumps

Addition to previous version of the response letter: Note a recent pre-print (Robel et al., 2021) finding similar transient dynamics to ours (so-called 'glacier persistence' at bed peaks) in the observational record and in simulations of glacier retreat over undulating topography. As in our study, this is explained with along-flow changes in longitudinal stress gradients.

Rewrote section 4.2 and inserted references to this section at appropriate locations in the text (c.f. previous answers)

*Figure 6 & 7 and related discussion: Isn't this simply showing mass conservation? For lateral variations, plotting Q/S is a proxy for the width-averaged velocity, which must increase where the geometry narrows simply due to mass conservation arguments.*

Mass conservation will indeed play a role in the retreat dynamics, but it is far from the full story. First, if Fig. 6 in the manuscript was showing mass conservation, the highest (lowest) width-averaged velocity would occur where S is minimized (maximised), not dS. Crucially, using dS rather than S as the predictive variable (x-axis in Fig. 6), clearly supports our interpretation that the *along-flow change* in fjord geometry, not only the absolute depth or width, is an important control on grounding line retreat. Therefore, we believe that Fig. 6 is not simply showing mass conservation. To further cement this claim, we plotted $v_{GL}$ over $S$ (Fig. S2). Mass conservation would predict a strong relationship with a high (low) $v_{GL}$ where $S$ is small (large). This is because a glacier needs to speed up (slow down) when the fjord is narrow (wide) to maintain the same ice flux. Figure S2 shows that such a relationship does not exist. In fact, there is a weak tendency towards smaller $v_{GL}$ for higher $S$, but this is not enough to explain the retreat dynamics that we observe.

Second, Q/S is not necessarily the width-averaged velocity, as is described in line 306ff: "Also, note that the GL flux is the product of the velocity $v_{GL}$ and the flux gate area at the GL $A_{GL}$, that is $Q_{GL} = v_{GL} \times A_{GL}$. The ratio $Q_{GL}/S$ is thus proportional to $v_{GL}$ when there is hydrostatic equilibrium at the GL (because in that case, $S = 0.9 \times A_{GL}$), ...". When the glacier front is grounded, there is not necessarily hydrostatic equilibrium at the GL, and then $Q_{GL}/S$ is not directly linked to $v_{GL}$. Finally, Fig. 7 shows the grounding line retreat rate $dGL$ plotted over the wetted area $S$. It is not obvious to us why that would be related to mass conservation in a straightforward way.

no changes to the text

*As correctly stated in equation (3), with this choice the integrated accumulation at the grounding line depends on the ice thickness at the inflow and the prescribed parameters only, i.e., is constant over most of the domain. The width-averaged ice flux at the grounding line at the beginning and the end of the transient (when presumably a steady state is attained) should thus only differ due to differences in ice thickness at the inflow boundary. For transient model results the picture is less clear, but the dynamically interesting quantity is the width-integrated grounding line flux. Does that show deviations from expected steady-state results (ideally in simulations without the elevation-accumulation feedback)?*

We are not entirely sure what the reviewer refers to here. The grounding line flux does indeed show deviations from steady-state values; in fact, this is a precondition for any retreat of any glacier where the surface mass balance does not change significantly. It is not clear to us why the grounding line flux would be expected to not do so, as is implied in the question. Unfortunately, we can not present any results without the elevation-accumulation feedback. However, we can show that the grounding line flux $Q_{GL}$ varies considerably as the grounding line retreats (see Fig. S3 below). These variations are much larger than the decrease in influx at the upstream domain boundary (max. $\sim 0.7 \times 10^9$ m$^3$ yr$^{-1}$, corresponding to 11% of total mass gains, c.f. answer to first comment). Hence it should be clear that

[Figure]

Figure S2: Relationship between velocity at the grounding line and the wetted area S for all geometries tested. For color code, please refer to Fig. 6 in the manuscript.

[Figure]

Figure S3: Grounding line discharge over time for different fjord geometries and perturbation magnitudes (small, medium, large). For color codes, see Fig. 2 in the manuscript. From top to bottom: linear reference fjord, embayments, depressions, bottlenecks, bumps. Note that simulations are stopped when a glacier has retreated to $x = 30$km; steady-state values are hence not obtained for all geometries at the end of the simulations.

the large fluctuations in grounding line flux occur due to the variable fjord topography.

no changes to the text

*Figure 8 & discussion: need to plot Q/S against dS to show that relationship still holds.*

As is stated in the manuscript (line 329): "Plotting all available data points for $Q_{GL}$/S over dS at Jakobshavn, we do not find the aforementioned geometric relationship." In the original manuscript, we do not claim that the relationship $Q_{GL}$/S over dS (as shown in Fig. 6 for our idealized fjords) holds in a quantitative way for Jakobshavn. Therefore, we do not see the need include a plot of Q/S against dS. We do mention, however, that the relationship holds in a qualitative way, "such that an increase in dS is generally associated with a decrease in $Q_{GL}/S$ and vice versa" (line 338). This 'qualitative' support is shown in Fig. 8e,d. In response to the reviewer, we will rephrase this paragraph to make it

clearer in a revised manuscript.

Added in line 350f: Even though this relationship is only qualitative, meaning that one value for $dS$ is not uniquely associated with one value for $Q_{GL}/S$, we find these results encouraging given the complexity of Jakobshavn Isbræ's dynamics

**References**

Aschwanden A, Fahnestock MA, Truffer M, Brinkerhoff DJ, Hock R, Khroulev C, Mottram R, Khan SA. 2019. Contribution of the Greenland Ice Sheet to sea level over the next millennium. Science Advances 5. doi:10.1126/sciadv.aav9396.

Boers N, Rypdal M. 2021. Critical slowing down suggests that the western Greenland Ice Sheet is close to a tipping point. Proceedings of the National Academy of Sciences 118. Publisher: National Acad Sciences.

Brinkerhoff D, Truffer M, Aschwanden A. 2017. Sediment transport drives tidewater glacier periodicity. Nature Communications 8:1–8. doi:10.1038/s41467-017-00095-5.

Brondex J, Gagliardini O, Gillet-Chaulet F, Durand G. 2017. Sensitivity of grounding line dynamics to the choice of the friction law. Journal of Glaciology 63:854–866. doi:10.1017/jog.2017.51.

Catania GA, Stearns LA, Sutherland DA, Fried MJ, Bartholomaus TC, Morlighem M, Shroyer E, Nash J. 2018. Geometric Controls on Tidewater Glacier Retreat in Central Western Greenland. Journal of Geophysical Research: Earth Surface 123:2024–2038. doi:https://doi.org/10.1029/2017JF004499.

Choi Y, Morlighem M, Wood M, Bondzio JH. 2018. Comparison of four calving laws to model Greenland outlet glaciers. The Cryosphere 12:3735–3746. doi:https://doi.org/10.5194/tc-12-3735-2018.

De Marco A, Mancini S, Miranda S, Scognamiglio R, Vitiello L. 2017. Experimental and numerical hydrodynamic analysis of a stepped planing hull. Applied Ocean Research 64:135–154. doi:10.1016/j.apor.2017.02.004.

Favier L, Gagliardini O, Durand G, Zwinger T. 2012. A three-dimensional full Stokes model of the grounding line dynamics: effect of a pinning point beneath the ice shelf. The Cryosphere :12.

Goldberg D, Holland DM, Schoof C. 2009. Grounding line movement and ice shelf buttressing in marine ice sheets. Journal of Geophysical Research: Earth Surface 114. doi:10.1029/2008JF001227.

Harrison WD, Elsberg DH, Echelmeyer KA, Krimmel RM. 2001. On the characterization of glacier response by a single time-scale. Journal of Glaciology 47:659–664. doi:10.3189/172756501781831837.

Haseloff M, Sergienko OV. 2018. The effect of buttressing on grounding line dynamics. Journal of Glaciology 64:417–431.

Khazendar A, Fenty IG, Carroll D, Gardner A, Lee CM, Fukumori I, Wang O, Zhang H, Seroussi H, Moller D, Noël BPY, van den Broeke MR, Dinardo S, Willis J. 2019. Interruption of two decades of Jakobshavn Isbrae acceleration and thinning as regional ocean cools. Nature Geoscience 12:277–283. doi:10.1038/s41561-019-0329-3.

Morlighem M, Wood M, Seroussi H, Choi Y, Rignot E. 2019. Modeling the response of northwest Greenland to enhanced ocean thermal forcing and subglacial discharge. The Cryosphere 13:723–734. doi:10.5194/tc-13-723-2019.

Robel A, Pegler S, Catania G, Felikson D, Simkins L. 2021. Ambiguous stability of glaciers at bed peaks. Geophysical Research Letters doi:10.1002/essoar.10507832.1. Preprint.

Schoof C, Davis AD, Popa TV. 2017. Boundary layer models for calving marine outlet glaciers. The Cryosphere 11:2283–2303.

Åkesson H, Morlighem M, O'Regan M, Jakobsson M. 2021. Future projections of Petermann Glacier under ocean warming depend strongly on friction law. Journal of Geophysical Research: Earth Surface .

Åkesson H, Nisancioglu KH, Giesen RH, Morlighem M. 2017. Simulating the evolution of Hardangerjøkulen ice cap in southern Norway since the mid-Holocene and its sensitivity to climate change. The Cryosphere 11:281–302. doi:10.5194/tc-11-281-2017.

Åkesson H, Nisancioglu KH, Nick FM. 2018. Impact of Fjord Geometry on Grounding Line Stability. Frontiers in Earth Science 6. doi:10.3389/feart.2018.00071.

---

## Author Response (AR2)

**Geometric Controls of Fjord Glacier Dynamics - response to reviewers II**

Frank, T., Åkesson, H., de Fleurian, B., Morlighem, M., Nisancioglu, K.H.

November 2021

**1 Response to Anonymous Referee #2**

We would like to thank the reviewer again for taking the time to review our manuscript a second time. We reply below by highlighting the reviewer's comments in italics above our answers. Changes made to the text are indicated under each of our answers in blue.

*I think it should be pointed out more clearly that the expected fast retreat through a widening fjord only happens after an initial period of almost-stagnant retreat at the head of the embayment, i.e., clearly marking the two different phases seen in the numerical runs (similarly for the other cases).*

Indeed, this is a very important aspect of our results! However, we feel that this point is already made in various places in the text. In fact, our entire analysis is centered around the question of why we see the stagnation before fast retreat occurs. For instance, sec. 3.1, 3.2 and 3.3 deal with exactly that question. Also in sec. 3.4, a specific reference is made in line 326. Furthermore, this is pointed out in the discussion in a paragraph dedicated to this point (line 394 - 402) as well as in line 410f and 425f. We therefore do not think it is necessary to change the text in this regard.

*Instead of focusing on the somewhat arbitrary definitions of S and dS (whose equations should be explicitly stated), the authors should focus on the physics. In particular, I think what they observe is that widening at the head of the embayment → leads to thinning → reduces velocity in vicinity of GL & GL discharge → slows down retreat (similar arguments can be made for the other cases).*
*Note that these effects are non-local due to the elliptic form of the SSA equations. Some of this is hinted at in line 427 "Essentially, we thus describe a mass conservation mechanism where a smaller flux gate requires a smaller ice flux to maintain the same grounding line position." but I think the manuscript would be strengthened considerably if the authors made more effort to test this hypothesis (looking at figure S3 seems to confirm this though but the supplemental itself was not accessible to me).*

With the brevity of the comment, it is not 100% clear what the reviewer refers to. But we will dissect the comment below as we understand it and respond to it accordingly.

1. "widening at the head of the embayment": In our understanding, head refers to the upstream side of the embayment and this is also what we believe makes more sense in the context of this comment. However, in the previous comment, head of the embayment referred to the downstream end. So we are not entirely sure what is meant here, but we will assume that the upstream end of the embayment is described.

2. "widening of fjord leads to thinning leads to reduced velocities and reduced GL discharge leads to slower retreat" (paraphrased from above): This chain of reasoning is, as we understand it, not consistent with our simulations. Indeed, a widening fjord causes thinning and reduced velocities but not reduced GL discharge, because the reduction in flow velocities is compensated by the wider flux gate. So GL discharge remains the same regardless of the width or depth of the fjord, unless another process (which, as elaborated in the manuscript, we believe is related to lateral and basal friction with the fjord) plays a role. If we interpret the comment correctly, the reviewer argues that a mass conservation argument is enough to describe the dynamics observed in our study, a point already made in the first round of review. However, as we discuss here, and as we have pointed out more extensively in the first response letter, this is not sufficient to explain our results.

3. comment on line 427: In light of the above remarks, we understand that this line may be confusing. We therefore suggest to remove it.

*It would be better to talk about stress gradients instead of dS, dS is just a metric to measure these. Similarly, it would be better to talk about cross-section or contact area, rather than S.*
In the context of our experimental setup and the interpretation of the results, dS is by no means just a metric for stress gradients. In fact, dS is a geometrical property of the fjord, while stress gradients are a dynamic response of the glacier to various external drivers, one being fjord geometry. So rather than dS being a metric for stress gradients, dS is the *cause* of observed stress gradients. Therefore, we do not see how it is meaningful to regard the two as some sort of synonym. Regarding the wetted area S, we have elaborated in the previous response letter on why it is meaningful to define such a quantity in the context of our study, and what the differences are compared to the cross-sectional area of the fjord.

*I think the authors should make more effort to clearly link their analysis to previous work: by focusing solely on flux per wetter area, this opportunity is missed, as they point out themselves: "glacier retreating on a prograde bed will experience a reducing wetted area as it recedes, and thus the ratio $Q_{GL}/S$ may increase, but not the absolute grounding line flux". I suspect that most readers will not have an intuitive grasp of $Q_{GL}/S$.*

We do not fully understand how the quote from the text cited here points towards a missed opportunity of putting our work in the context of previous studies. However, we agree with the reviewer that most readers will not have an intuitive understanding of $Q_{GL}/S$. Therefore, we suggest to add some further explanation in the text in line 391: However, the physical interpretation of the relationship between $Q_{GL}/S$ and $dS$ is not straight-forward. Since $Q_{GL}/S$ is proportional to $v_{GL}$ when there is hydrostatic equilibrium at the grounding line, the expression can be thought of as relating grounding line velocities to along-fjord changes in fjord topography, through the mechanisms described in our stress analysis (sec. 3.3). Accordingly, our results show that velocity evolution at the grounding line over time is also a good proxy for the dynamic response of a glacier to fjord topography (Fig. SA4). This may be specifically useful for less well-studied glaciers with unknown bathymetry. Notably, our geometric relationship is distinct from a typical mass-conservation argument, which simply states that velocities must increase for a decreased flux gate, and vice versa, to maintain the same grounding line discharge. This is because in such an argument, velocities are related to absolute values of fjord width or depth, and not the along-fjord change of fjord geometry. While mass-conservation mechanisms certainly play a role in our simulations, we do not find that such a relationship alone is sufficient to fully explain the dynamics we observe.

---

## Author Response (AR3)

**Geometric Controls of Fjord Glacier Dynamics - response to reviewers III**

Frank, T., Åkesson, H., de Fleurian, B., Morlighem, M., Nisancioglu, K.H.

November 2021

**1 Response to editor**

We would like to thank the editor for his additional remarks to improve our manuscript. We reply below by highlighting the editor's comments in italics above our answers. Changes made to the text are indicated under each of our answers in blue.

*L144, 147 Please define S and dS in equation form.*

Done.

The metric used to quantitatively link fjord shape with glacier retreat is the wetted area $S$: the submerged cross-sectional area of the fjord (Fig. 1b), which can be calculated at any point along an outlet channel according to

$$S(x') = \int_{y_1'}^{y_2'} D(x', y')dy' \tag{1}$$

where $D$ is the water depth, $y_1'$ and $y_2'$ are the intersections of the water line in the fjord with the fjord walls so that $y_2' - y_1'$ is the width of the outlet channel in a transformed coordinate system that is oriented such that the coordinates $(x', y')$ are parallel and perpendicular to the center line of the outlet channel, respectively, at any given point. $S$ combines information about the width and depth of a fjord and is thus a comprehensible parameter for comparing basal and lateral perturbations. Furthermore, it is straightforward to calculate its first derivative $\frac{dS}{dx'}$ (in the following written as $dS$ for brevity) which yields information on the along-fjord change in width and/or depth.

*L163, L180 Please consider using 'steady' instead of 'stable' (perhaps also a pet peeve of mine, and previously raised by reviewer 2, but stability should only be used when referring to the system-response of a steady state to small perturbations)*

Done.

*If the linear relationship between qGL/S and dS is indeed 'universal', and therefore applies to all laterally-confined glacier systems including the area of Jakobshavn identified in Fig 8c, why not overlay the data points from Fig 8d/8e on top of the results from idealized experiments in Fig 6?*

As we state in the manuscript L348f, the relationship that we see for JI is of qualitative nature, "such that an increase in $dS$ is generally associated with a decrease in $Q_{GL}/S$ and vice versa". So the

sign of the change in $dS$ and $Q_{GL}/S$ is inversely correlated for JI, not the actual values. Therefore, it would not be meaningful to plot the data points from Fig 8d/e on the same plot as the results from the idealized experiments - no visual correlation would appear.

We understand that we have not been clear enough in the usage of the word 'universal', i.e. whether we refer to it in the context of our idealized experiments only or also regarding the results for JI. We therefore suggest to remove the word universal from the section name 3.4, and make changes to the text as specified below to make it clear that the relationship is universal in the context of our idealized experiments, but not quantitatively universal to be applied directly to the results for JI.

L382f: Given these disparities between different settings, it is all the more compelling that we find the geometric relationship $Q_{GL}/S$ over $dS$ universal to all our idealized fjords.

L385: To the authors knowledge, this type of quantitative link between fjord topography and glacier response has not been established before,...

*L11, L76. For clarity, consider replacing 'long-term study of the retreat' by '???-year simulation of the Holocene retreat of' ?*
As given in L208f in the manuscript, the simulation of Jakobshavn Isbræ used here 'is a sensitivity experiment not meant to reflect the actual evolution of JI (Kajanto et al., 2020), [but] it is convenient for our purposes since it produces a long-lasting, dynamic retreat.' Since it therefore does not reflect the actual retreat of JI in the Holocene, we suggest to refrain from using the word 'Holocene' in L11 and L76 as it might give a false impression on the reader. However, we will add that the run simulates 8 kyr in section 2.5 'A real-world case study: Jakobshavn Isbrae'.

We focus on a 8000-year simulation of the retreat of JI from a sill at the fjord mouth of Jakobshavn Isfjord, to a point inland of today's GL position.

*L87 Since spatial variations in shear stress form an important element of this study, readers less familiar with the parameterizations might like to see more detail on how viscosity is calculated in the model. E.g. can you provide an equation, which explicitly shows the dependency of the viscosity on the friction?*

The phrase 'feedback between frictional heating and ice viscosity' simply refers to the fact that we use a thermal model to constrain ice viscosity, so 'frictional heating' here refers to the frictional heating between the ice crystals. Besides that, there is no specific feedback associated with friction between the ice and the fjord walls. We therefore suggest to rephrase and refer to Larour et al. (2012) where more details on the thermal model are given.
We use a thermal model (Larour et al., 2012) to constrain the ice rheology parameter, B, based on the temperature dependent relationship from Cuffey and Paterson (2010).

*L93 Say 'Ice front retreat' instead of 'calving'? If $\sigma = \sigma_{max}$, the calving rate is equal to the ice velocity according to (1) and the ice front does not move.*
Done.

*L113 The description of the melt parameterization might need further clarification and/or an equation to describe its behavior. My understanding is that 30m/yr melt is applied to all floating nodes, and 200 m/yr to nodes along the ice front (effectively describing a fixed calving rate), is that correct? Also, do you use any of the sub-grid melt schemes in ISSM, which have been shown to influence GL dynamics?*

ISSM accounts for mass loss through ocean induced melt in two different ways: 1) A melt rate (in this study 30 m/yr for the reference glacier) is applied to all floating parts of the glacier from

below, thus thinning floating ice. Since ISSM assumes a direct connectivity between the ocean and the subglacial hydrology (not only for ocean melt rates, but also when calculating the effective pressure in the description of basal friction), this also applies to the elements in the basal cavities referred to in the last comment below. 2) A frontal rate of undercutting is applied to elements at the ice front if they are grounded, and thus effectively acts as an extra retreat rate. This parameterization in ISSM is grounded in the theory of ocean circulation in response to freshwater plumes at the grounding line, and experimentally derived using ocean model output (Rignot et al., 2016). The value chosen for this parameter encapsulates both the small scale calving events resulting from undercutting as well as the direct melt at the ice front. Morlighem et al. (2019) use salinity and water temperature outputs from ocean modelling to calculate the frontal rate of undercutting for glaciers in northwest Greenland, while we prescribe them as part of our idealized setup (using a conservative estimate of 200m/yr for the reference glacier).

We do use the sub-element melt scheme 'full melt on partially floating'. We have conducted experiments which indicate that the mesh is sufficiently refined in the vicinity of the grounding line that the type of subelement scheme for both friction and melt does not affect the simulations significantly.

L112ff: Ocean induced melting is parameterized through prescribed melt rates that are invariant of depth. On all elements that have a floating section, a fixed basal melt rate is applied, thinning the ice from below. The reference forcing for this subshelf melt rate used for model spin-up is 30 m $\mathrm{yr}^{-1}$. All elements at the ice front are subject to a frontal rate of undercutting if they are grounded. This parameterization accounts for both small scale calving events associated with undercutting and direct melt at the ice front (Rignot et al., 2016; Morlighem et al., 2019). The reference forcing here is 200 m $\mathrm{yr}^{-1}$. Both values are on the lower end of observed melt rates (Motyka et al., 2003; Enderlin and Howat, 2013; Xu et al., 2013), thus reflecting a configuration prior to recent warming when glaciers were more in balance with the ambient climate than today (King et al., 2020). For both friction and melt, experiments not shown here indicate that the mesh is sufficiently refined in the vicinity of the grounding line that the type of subelement scheme chosen does not affect the simulations significantly.

L149 Refer to Fig2 here as well as Table2
Done.

L168 'GL position remains largely unchanged' instead of 'stable'
Done.

L223 Do you apply basal melt in these cavities? I assume not, since they are not connected to the open ocean? Please specify as part of the description in L113 (see comment above).

see above

**References**

Cuffey KM, Paterson WSB. 2010. The physics of glaciers. Amsterdam: Butterworth-Heinemann.

Enderlin EM, Howat IM. 2013. Submarine melt rate estimates for floating termini of Greenland outlet glaciers (2000–2010). Journal of Glaciology 59:67–75. doi:10.3189/2013JoG12J049.

King MD, Howat IM, Candela SG, Noh MJ, Jeong S, Noël BPY, van den Broeke MR, Wouters B, Negrete A. 2020. Dynamic ice loss from the Greenland Ice Sheet driven by sustained glacier retreat. Communications Earth & Environment 1:1–7. doi:10.1038/s43247-020-0001-2.

Larour E, Seroussi H, Morlighem M, Rignot E. 2012. Continental scale, high order, high spatial resolution, ice sheet modeling using the Ice Sheet System Model (ISSM). Journal of Geophysical Research: Earth Surface 117. doi:10.1029/2011JF002140.

Morlighem M, Wood M, Seroussi H, Choi Y, Rignot E. 2019. Modeling the response of northwest Greenland to enhanced ocean thermal forcing and subglacial discharge. The Cryosphere 13:723–734. doi:10.5194/tc-13-723-2019.

Motyka RJ, Hunter L, Echelmeyer KA, Connor C. 2003. Submarine melting at the terminus of a temperate tidewater glacier, LeConte Glacier, Alaska, U.S.A. Annals of Glaciology 36:57–65. doi:10.3189/172756403781816374.

Rignot E, Xu Y, Menemenlis D, Mouginot J, Scheuchl B, Li X, Morlighem M, Seroussi H, Broeke Mvd, Fenty I, Cai C, An L, Fleurian Bd. 2016. Modeling of ocean-induced ice melt rates of five west Greenland glaciers over the past two decades. Geophysical Research Letters 43:6374–6382. doi:10.1002/2016GL068784.

Xu Y, Rignot E, Fenty I, Menemenlis D, Flexas MM. 2013. Subaqueous melting of Store Glacier, west Greenland from three-dimensional, high-resolution numerical modeling and ocean observations. Geophysical Research Letters 40:4648–4653. doi:10.1002/grl.50825.